# Decouple and Cache: KV Cache Construction for Streaming Video Understanding

**Zhanzhong Pang** [1]  **Dibyadip Chatterjee** [1]  **Fadime Sener**  **Angela Yao** [1]

https://github.com/pangzhan27/DSCache

## Abstract

Streaming video understanding requires processing unbounded video streams with limited memory and computation, posing two key challenges. First, continuously constructing new and evicting old key-value (KV) caches is required for unbounded streams. Secondly, due to the high cost of collecting and training on unbounded streams, models must learn from short sequences while generalizing to long streams. Existing streaming VideoVLLMs fail to scale to unbounded video streams or focus on cache reuse strategies, leaving the impact of cache construction underexplored. In this paper, we propose Decoupled Streaming Cache (DSCache), a training-free cache construction mechanism that adapts pretrained offline models to streaming settings. DSCache maintains a cumulative past KV cache while constructing a separate instant cache on-demand, decoupled from past caches to preserve the informativeness of recent inputs. To enable position extrapolation beyond the training length, DSCache further incorporates a position-agnostic encoding strategy, ensuring KV caches to support unseen positions and preventing position overflow. Experiments on Streaming Video QA benchmarks demonstrate DSCache's state-of-the-art performance, with an average 2.5% accuracy gains over prior methods.

## 1. Introduction

Streaming video understanding is central to perception in robotics, surveillance, and autonomous driving (Supancic III & Ramanan, 2017; Muhammad et al., 2019; Wei et al., 2022; Shao et al., 2024; Pang et al., 2025). However, it remains highly challenging, as models must process continuous, unbounded video streams and make real-time predictions from incomplete observations. Many deployed applications also operate under strict memory and computational constraints. Existing streaming VideoLLMs (Zhang et al., 2024; Yao et al., 2025; Fu et al., 2025; Li et al., 2025) adopt proactive and asynchronous LLM invocation designs, but most remain quasi-streaming, as they process fixed, pre-defined segments and lack explicit designs that scale to unbounded streams.

Real unbounded streaming introduces two major challenges. First, streaming inference requires maintaining KV caches to preserve past context for efficient prediction. Under unbounded streams, these caches must be continuously updated and evicted to meet resource limits. Existing methods (Li et al., 2024b; Xu et al., 2025; Ning et al., 2025) adopt *uniform KV cache construction*, updating caches in the same manner for each incoming input, with older caches evicted, compressed, or offloaded and later retrieved (Zhang et al., 2023; Di et al., 2025; Kim et al., 2026). While these approaches prioritize efficient cache utilization, the quality and reliability of constructed caches remain underexplored. Second, models are typically trained on short sequences due to the high cost of collecting and training on unbounded data, yet are expected to generalize to long streams at inference time (Press et al., 2021; Yang et al., 2025). To align inference behavior with training, prior works (Xiao et al., 2024; Xu et al., 2025) introduced attention sinks to stabilize streaming inference after cache eviction, and addressed position extrapolation beyond the training length, relying on empirical strategies and lacking formal analysis.

In this paper, we adapt pretrained offline models for Streaming Video Question Answering (StreamingVQA) in a training-free setting, and investigate key factors that limit such adaptation. As shown in Figure 1, a naive approach directly processes sampled frames at each time step (Yao et al., 2025; Pang et al., 2026), incurring redundant recomputation, higher latency, and limited temporal context. Alternatively, offline models can maintain a uniform KV cache (Xiao et al., 2024), where each new cache is updated in the same manner based on past caches. This avoids costly recomputation and enables long-context utilization. However, under

[1]School of Computing, National University of Singapore, Singapore. Correspondence to: Zhanzhong Pang <pang@comp.nus.edu.sg>.

*Proceedings of the 43rd International Conference on Machine Learning*, Seoul, South Korea. PMLR 306, 2026. Copyright 2026 by the author(s).

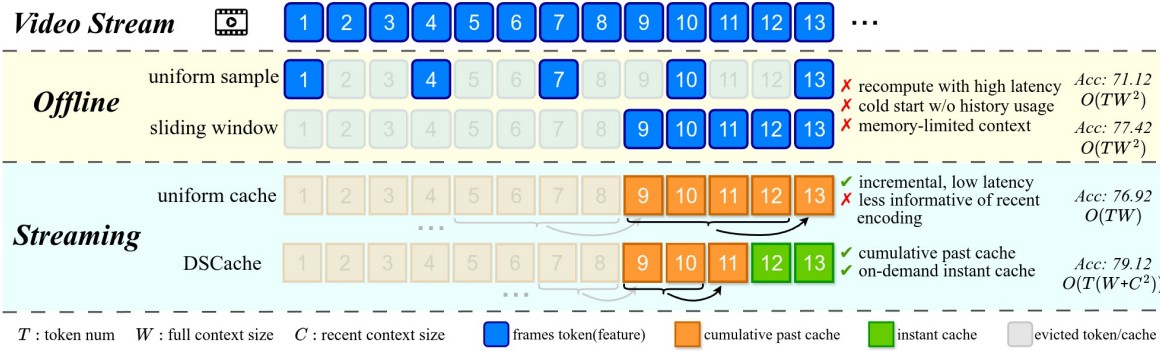

*Figure 1.* Illustration of DSCache vs. existing methods. Inference over a video stream of length $T$ with a full context of size $W$ and a separate recent context of size $C \ll W$. Offline models sample frames from the past, whereas the streaming baseline uniformly updates the KV cache while evicting older ones. DSCache decouples the cumulative past and instant cache constructions to better preserve fine-grained recent information. Accuracy is reported on StreamingBench using Llava-OV-7B.

uniform KV cache construction, we observe a cumulative effect in which new caches are built on past caches that retain residual information from earlier, evicted ones. This issue arises in the presence of cache eviction and depends on how caches are constructed and encoded. Since pretrained models exhibit strong recency biases (Liu et al., 2023; Tian et al., 2025), this indirect accumulation causes new caches to retain excessive historical information, reducing their informativeness for current inputs (see Supplementary for further analysis). In addition, as inputs are processed incrementally, token positions eventually exceeds those seen during training, resulting in out-of-distribution and overflow. Cache eviction further introduces discontinuities in positions between visual and system tokens, violating the continuity assumption made during training. Enabling KV caching at out-of-distribution positions while maintaining continuity in positions is critical for unbounded streaming.

Based on these observations, we propose a training-free, cache construction mechanism, Decoupled Streaming Cache (DSCache), to mitigate the cumulative effect during cache construction, and to support unbounded streams by handling discontinuous and out-of-distribution positions beyond those seen during training. Specifically, DSCache maintains a cumulative past KV cache while constructing a separate instant cache on demand for incoming inputs. Using a feature buffer, it encodes recent inputs without cumulative interference from the past KV cache, thereby preserving their informativeness. We further integrate DSCache with a position-agnostic encoding strategy. Unlike standard position-encoded KV caches, this strategy stores caches without positional information and assigns them only at usage time, allowing models to reinitialize positions, thereby handling positions that are discontinuous or beyond training lengths, preventing position overflow. We formally prove that, for RoPE-based LLMs, position-agnostic encoding with reassigned positions produces outputs equivalent to standard positional encoding. This guaran-

tees correctness while supporting flexible cache operations, e.g. partitioning, disassembling, and selective reuse.

We evaluate DSCache on StreamingVQA, which requires real-time understanding of continuous video streams to answer user queries. DSCache achieves state-of-the-art performance, with an average 2.5% accuracy improvement over existing methods on standard benchmarks (Lin et al., 2024; Niu et al., 2025), while enabling stable memory utilization and low-latency inference. Summarizing our contributions,

- We propose a training-free method that adapts offline pretrained models to unbounded streaming.
- We reveal a cumulative effect in uniform streaming KV caches, where residuals from earlier caches degrade the quality of encoding recent inputs.
- We introduce DSCache, separating cumulative and instant caches to preserve informativeness of recent inputs while maintaining accumulated context.
- We present a position-agnostic encoding strategy with a formal equivalence proof, enabling position extrapolation beyond training lengths and supporting flexible cache operations.

## 2. Related works

**Streaming VideoLLMs** extend large multi-modal models to process continuous video streams and support timely interactions and responses. VideoLLM-online (Chen et al., 2024a) first introduced proactive online video interaction by invoking LLMs at every step. Subsequent works, StreamMind (Ding et al., 2025) and StreamBridge (Wang et al., 2026), improved efficiency by selectively triggering the LLMs only on relevant events. Dispider (Qian et al., 2025) and ViSpeak (Fu et al., 2025) further enabled asynchronous processing by decoupling perception and response generation. Complementary works, such as Timechat-online (Yao et al., 2025), improved efficiency by reducing visual redundancy through token filtering (Li et al., 2025), compres-

sion (Zhang et al., 2024), or adaptive selection (Wu et al., 2024). These models typically emphasize responsiveness and efficiency, and are evaluated under quasi-streaming setup, processing discrete segments without scalable KV cache design. In contrast, we focus on unbounded streaming inference with continuous KV cache management.

**Streaming Inference** shifts the focus from model-level behavior to inference length generalization and cache management. Early works, such as StreamingLLM (Xiao et al., 2024) and LM-Infinite (Han et al., 2024), investigated attention bottlenecks and provided theoretical insights for infinite-length inference without finetuning. LongLoRA (Chen et al., 2024b) adapted LLMs to long sequences via sparse local attention and LoRA fine-tuning. StreamingVLM (Xu et al., 2025) extended StreamingLLM to video, reusing KV caches of sinks, visual and text tokens. To preserve long-range dependencies and reduce redundancy, recent works focus on improving KV cache management. ProVideLLM (Chatterjee et al., 2025) introduced an interleaved multimodal cache, using verbalized text caches to retain long-range visual information. ReKV (Di et al., 2025) stored processed KV caches in RAM and dynamically retrieving query-relevant entries. Other training-free approaches applied cache pruning or compression based on user queries (Ning et al., 2025) or attention scores (Kim et al., 2026; Yang et al., 2025) to improve efficiency. However, existing methods assume a standard, uniform KV cache construction, and ignore how the construction quality affects downstream performance. Our method fills the gap by focusing on cache construction, enhancing cache quality to improve inference performance.

# 3. Preliminaries

We consider unbounded streaming inference, where a model incrementally processes video streams and answers user queries solely based on observed information under resource constraints. Since collecting data and training in this unbounded regime is costly, we adopt a *training-free* setup that adapts a pretrained offline model for streaming inference. To avoid redundant computation, the model maintains a compact key-value cache (KV cache) that summarizes the past context, rather than recomputing over sliding windows at each time step.

## 3.1. Streaming Inference Scheme

At each time step $t$, the model receives current input tokens $X_t = \{x_{i+l_P}\}_{i=0}^{l_X-1}$, which include newly arriving frame tokens and any posed question tokens, with the past KV cache $\mathcal{P} = \{\mathcal{P}^{(r)}\}_{r=0}^{N-1}$, $|\mathcal{P}^{(r)}| = l_P$. Here, $N$ denotes the number of transformer layers, $l_P$ the total history length processed up to $t$, and $l_X$ the input sequence length. Due to memory constraints, the past KV cache is truncated to a fixed context window of length $l_W \leq l_P$. In addition, atten-

tion sinks (Xiao et al., 2024), where early tokens disproportionately attract high attention, can dominate the model's focus in streaming inference. To mitigate their effect, the KV cache for the first $l_A$ tokens is maintained separately as $\mathcal{C}_a = \{\mathcal{C}_a^{(r)}\}_{r=0}^{N-1} = \{\mathcal{P}_{0:l_A}^{(r)}\}_{r=0}^{N-1}$. The KV cache used at time $t$ is then $\mathcal{C}_t = \{[\mathcal{C}_a^{(r)}, \mathcal{P}_{l_P-l_W:l_P}^{(r)}]\}_{r=0}^{N-1}$, where $[\cdot]$ denotes concatenation. Slicing and concatenation are applied independently to the key and value caches at each layer along the sequence dimension.

**Stream Encoding (Prefilling).** The model encodes input tokens while updating the corresponding KV cache, typically serving as a prefilling stage when the input includes a user question. Let $Z_t^{(r)}$ denote the input to the $r$th transformer layer with $Z_t^{(0)} = X_t$. For higher layers, the input is computed as $Z_t^{(r+1)} = g_r(Z_t^{(r)}, \mathcal{C}_t^{(r)})$ where $g_r$ denotes the computation of the $r$th layer, and $\mathcal{C}_t^{(r)} = (K_t^{(r)}, V_t^{(r)})$ is the KV cache at that layer $r$. During attention calculation, queries and keys are encoded with positional information. The resulting keys and the corresponding values are concatenated with the reused past KV cache $\mathcal{C}_t^{(r)}$, forming the updated KV cache for the layer.

$$Q_{t+1}^{(r)} = f(W_q^{(r)} Z_t^{(r)}, l_P), K_{t+1}^{(r)} = [K_t^{(r)}, f(W_k^{(r)} Z_t^{(r)}, l_P)],$$
$$V_{t+1}^{(r)} = [V_t^{(r)}, W_v^{(r)} Z_t^{(r)}] \tag{1}$$

where $f(\cdot, m)$ applies positional encoding using contiguous position IDs starting from $m$. $W_q^{(r)}$, $W_k^{(r)}$, and $W_v^{(r)}$ are the parameters of the $r$th layer.

After encoding, the past KV cache $\mathcal{P}$ is updated to length $l_P + l_X$ to include the newly added entries. At the next time step $t + 1$, the reused cache $\mathcal{C}_{t+1}$ is constructed by concatenating the attention sinks with the most recent $l_W$ tokens from the updated past cache, evicting older entries as needed, yielding $\mathcal{C}_{t+1} = \{[\mathcal{C}_a^{(r)}, \mathcal{P}_{l_P-l_W+l_X:l_P+l_X}^{(r)}]\}_{r=0}^{N-1}$.

**Decoding.** When the input includes a question, autoregressive decoding is performed using the KV cache $\mathcal{C}_{t+1}$ built incrementally in encoding stage to generate responses.

## 3.2. Position Re-indexing

Concatenating attention sinks with truncated past caches after eviction results in discontinuous position indices. In addition, position IDs grow linearly over time, leading to positions unseen during training and potential index overflow. These issues may cause generalization failures and out-of-distribution behavior (Chen et al., 2023). To stabilize unbounded streaming inference, position indices must therefore remain within a bounded range. However, in standard KV caches, keys are encoded with positional information during cache construction, as in Equation (1), which fixes positional transformations and prevents later modification. To address this limitation, prior works (Xiao et al., 2024;

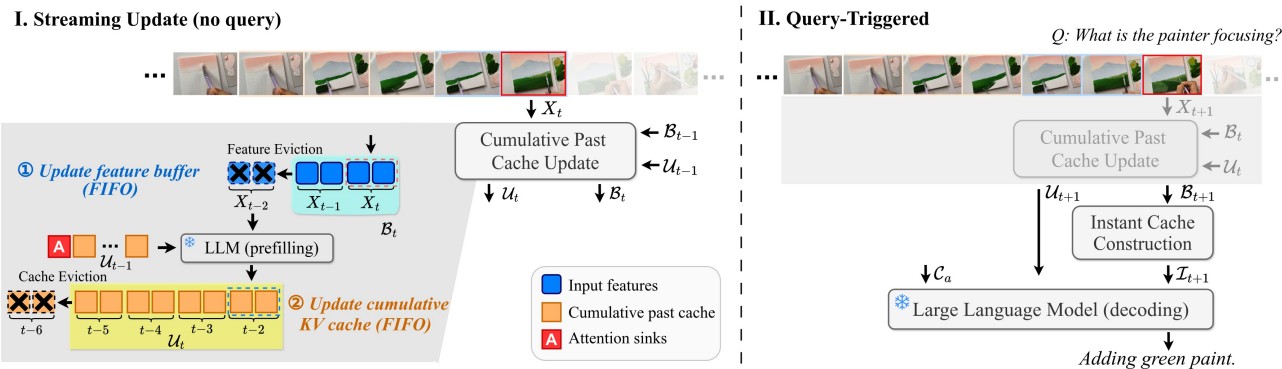

*Figure 2.* Streaming video inference with DSCache. DSCache maintains a feature buffer $\mathcal{B}$ that stores recent inputs $X$ in FIFO order, and a cumulative past KV cache $\mathcal{U}$. At each time step $t$, new features $X_t$ enter the buffer and older features $X_{t-2}$ are evicted to update the cumulative past KV cache $\mathcal{U}_t$. When inference is required at $t+1$, e.g. due to an incoming query, the buffer and the cumulative past KV cache are updated first. The instance cache $\mathcal{I}_{t+1}$ is then constructed from the buffer $\mathcal{B}_{t+1}$ alone to preserve detailed recent context, and then combined with the cumulative past cache $\mathcal{U}_{t+1}$ to generate the response.

Han et al., 2024) on RoPE-based LLMs store KV caches before positional transformations, allowing positional information to be applied at use time. Specifically, positional transformations are applied to the concatenation of reused and newly generated keys starting from position 0, and to queries derived from new inputs starting from $l_A + l_W$.

$$Q'^{(r)}_{t+1} = f(W_q^{(r)} Z_t^{(r)}, l_A + l_W),$$
$$K'^{(r)}_{t+1} = f([K'^{(r)}_t, W_k^{(r)} Z_t^{(r)}], 0). \quad (2)$$

This formulation enables modified position IDs, allowing positions to be shifted upon cache eviction while remaining contiguous, bounded, and in-distribution. However, this strategy is based on empirical heuristics and lacks formal proof of its correctness.

## 4. Methodology

We propose Decoupled Streaming Cache (DSCache), a mechanism for KV cache construction and maintenance in streaming settings. To enable position extrapolation beyond the training length, DSCache incorporates a position-agnostic encoding strategy, enabling flexible cache construction for unbounded streaming (Section 4.1). At each time step, DSCache updates a cumulative past KV cache based on the incoming inputs, aggregating historical context while maintaining a fixed memory budget via cache eviction (Section 4.2). When a user query arrives, an instant KV cache is created on demand to capture fine-grained details for inference, during which the model generates responses using both the cumulative and instant KV caches (Section 4.3). Importantly, DSCache operates solely at the KV cache construction stage and is orthogonal to existing cache utilization techniques, such as compression, pruning, or retrieval.

### 4.1. Position-Agnostic Encoding

To support unseen positions beyond training lengths and prevent position overflow, we introduce the position-agnostic encoding, which enables flexible position handling during cache construction for unbounded streaming inference.

**Definition 4.1.** Position-agnostic encoding stores a KV cache prior to applying positional transformations, requiring positional information to be reintroduced at use time.

We establish the correctness properties of position-agnostic encoding. Notably, the practical reassignment of contiguous position IDs to attention sinks and reused caches in Equation (2) slightly alters relative distances and is therefore not covered by the formal proposition.

**Proposition 4.2.** *For RoPE-based LLMs, position-agnostic encoding with shifted position IDs that preserve relative token distances produces outputs identical to standard positional encoding with absolute position IDs.*

**Corollary 4.3.** *Under position-agnostic encoding, cache entries can be partitioned, recombined, or selectively reused through reassignment of position IDs.*

The proof is provided in the Supplementary. These properties form the foundation for flexible cache utilization in streaming inference. We adopt position-agnostic encoding as the default formulation for the remainder of the paper.

### 4.2. Cumulative Past KV Cache Update

Standard streaming inference with uniform KV caches suffers from a cumulative effect, where evicted caches continue to influence the construction of newly added caches, degrading the informativeness of encoded recent inputs. To enhance the quality of recent input encoding while maintain accumulated past context, we decouple the storage of recent input features from cumulative KV cache update. Specifically, at time step $t$, DSCache maintains a feature buffer

$\mathcal{B}_{t-1}$ with fixed size $l_I$ that stores the most recent inputs. Upon arrival of the new input $X_t$, the buffer is updated in a first-in-first-out (FIFO) manner by appending $X_t$ and evicting the oldest entry $X_{t-l_I}$ when full:

$$\mathcal{B}_t = \begin{cases} [\mathcal{B}_{t-1}, \{X_t\}], & |\mathcal{B}_{t-1}| < l_I, \\ [\mathcal{B}_{t-1} \setminus \{X_{t-l_I}\}, \{X_t\}], & |\mathcal{B}_{t-1}| = l_I. \end{cases} \quad (3)$$

The evicted input $X_{t-l_I}$ is then used to update the cumulative past KV cache. To formalize this process, we introduce a KV cache construction operator $\varphi$, which generates KV caches for newly added tokens under the position-agnostic encoding. Given an input $X$ and past KV caches $\mathcal{C}$, let $Z^{(0)} = X$ and $Z^{(r+1)} = g_r(Z^{(r)}, \mathcal{C}^{(r)})$ denote the input of layer $r + 1$. The operator projects the input $Z^{(r)}$ of each layer into KV caches.

$$\varphi(X, \mathcal{C}) = \left\{ (W_k^{(r)} Z^{(r)}, \ W_v^{(r)} Z^{(r)}) \right\}_{r=0}^{N-1}, \quad (4)$$

where $\varphi$ produces KV caches only for newly added tokens and excludes previously stored caches.

Given the previous cumulative past KV cache $\mathcal{U}_{t-1}$, the attention sink KV cache $\mathcal{C}_a$, and evicted feature $X_{t-l_I}$, the cumulative KV cache is updated as

$$\mathcal{U}_t = \left[ \mathcal{U}_{t-1}, \ \varphi(X_{t-l_I}, [\mathcal{C}_a, \zeta(\mathcal{U}_{t-1})]) \right], \quad (5)$$

where $\zeta$ denotes a subsampling operator that selects a subset of the cumulative KV cache of length $l_L$. For the cumulative KV cache, we explicitly distinguish between the stored length $l_U$ and the active length $l_L$ used during cache construction, with $l_L \leq l_U$. This formulation retains an extended history while selectively conditioning cache updates on only a subset of past KV caches. In practice, we define $\zeta$ to select the most recent $l_L$ KV caches.

If the updated caches exceed the budget $l_U$, older tokens are evicted in FIFO order along the sequence-length axis, applied independently for each transformer layer $r$.

$$\mathcal{U}_t^{(r)} = \begin{cases} \mathcal{U}_t^{(r)}, & |\mathcal{U}_t^{(r)}| \leq l_U, \\ \{(K_i^{(r)}, V_i^{(r)})\}_{i=|\mathcal{U}_t^{(r)}|-l_U}^{|\mathcal{U}_t^{(r)}|-1}, & |\mathcal{U}_t^{(r)}| > l_U. \end{cases} \quad (6)$$

In addition, when the evicted input $X_{t-l_I}$ contains text (e.g. previous queries or responses), the corresponding text KV caches can be removed from $\mathcal{U}_t$ to avoid interference with subsequent visual caching. This is useful for settings where subsequent queries are independent of prior text.

### 4.3. KV Cache-based Inference

Building on the updated cumulative KV cache and the feature buffer, DSCache supports inference by constructing an instant KV cache from the buffer, which is then used jointly with the cumulative past KV cache.

When inference is required, e.g. upon a user query at $t+1$, after the cumulative past KV cache update, the instant cache is constructed solely from the current feature buffer $\mathcal{B}_{t+1}$, independent of the cumulative cache $\mathcal{U}_{t+1}$. During construction, the buffer inputs may be temporarily re-encoded, e.g. using a higher-resolution version of $X_{t+1}$, without modifying the buffer itself, ensuring consistent features for subsequent cumulative cache updates. The feature buffer preserves fine-grained recency information, ensuring that the instant KV cache construction remains isolated from the cumulative cache. To align with training, attention sink caches $\mathcal{C}_a$ are included during the instant cache construction:

$$\mathcal{I}_{t+1} = \varphi(\mathcal{B}_{t+1}, \mathcal{C}_a), \quad (7)$$

where $\varphi$ is applied to the buffer independently of $\mathcal{U}_{t+1}$.

The final KV cache for inference is formed by concatenating the attention sink, cumulative, and instant caches as

$$\mathcal{C}_{t+1} = [\mathcal{C}_a, \mathcal{U}_{t+1}, \mathcal{I}_{t+1}] \quad (8)$$

As in the cumulative cache construction, cache selection (e.g. subsampling or retrieval) may be applied before concatenation. After inference, the buffer can be augmented with the response to $X_{t+1}$ for reuse in subsequent steps. Admittedly, recomputing the instant cache incurs additional computation, but the overhead is minor for a small buffer. By decoupling cumulative and instant KV caches, DSCache balances efficiency and informativeness.

In all, DSCache involves three distinct attention computations at different stages of streaming, where positional information is applied at use time: the cumulative KV cache update, the instant cache construction, and inference using the final KV cache. For all three, we adopt RoPE-based LLMs with position-agnostic encoding and assign consecutive positional indices starting at zero to each entry in the active KV cache. With appropriate cumulative cache and feature buffer sizes, positional indices remain bounded during streaming inference. In addition, by assigning consecutive positional IDs across attention sinks and the cumulative KV cache, streaming inference mimics a contiguous training sequence, thereby mitigating the train–inference discrepancy.

## 5. Experiments

### 5.1. Experimental Setup

**Benchmarks.** We primarily evaluate our approach on multiple StreamingVQA benchmarks. StreamingBench (Lin et al., 2024) is widely adopted; we focus on the real-time visual understanding task, which contains 500 videos with 2500 QA-pairs. OVO-Bench (Niu et al., 2025) evaluates real-time perception along with complementary capabilities of backward tracing and forward active responding,

*Table 1.* Comparisons on StreamingBench Real-Time Visual Understanding task, evaluated by accuracy. Uniform cache denotes the baseline with uniform streaming KV caching. This task is organized into 10 subtasks: Object Perception (OP), Causal Reasoning (CR), Clips Summarization (CS), Attribute Perception (ATP), Event Understanding (EU), Text-Rich Understanding (TR), Prospective Reasoning (PR), Spatial Understanding (SU), Action Perception (ACP), and Counting (CT).

| Model | | #Frames | OP | CR | CS | ATP | EU | TR | PR | SU | ACP | CT | All |
|---|---|---|---|---|---|---|---|---|---|---|---|---|---|
| **Proprietary MLLMs** | | | | | | | | | | | | | |
| Gemini 1.5 pro (Team et al., 2024) | | 1 fps | 79.02 | 80.47 | 83.54 | 79.67 | 80.00 | 84.74 | 77.78 | 64.23 | 71.95 | 48.70 | 75.69 |
| GPT-4o | | 64 | 77.11 | 80.47 | 83.91 | 76.47 | 70.19 | 83.80 | 66.67 | 62.19 | 69.12 | 49.22 | 73.28 |
| **Open-source Online VideoLLMs** | | | | | | | | | | | | | |
| Flash-VStream-7B (Zhang et al., 2024) | | - | 25.89 | 43.57 | 24.91 | 23.87 | 27.33 | 13.08 | 18.52 | 25.20 | 23.87 | 48.70 | 23.23 |
| VideoLLM-online-8B (Chen et al., 2024a) | | 2 fps | 39.07 | 40.06 | 34.49 | 31.05 | 45.96 | 32.40 | 31.48 | 34.16 | 42.49 | 27.89 | 35.99 |
| Dispider-7B (Qian et al., 2025) | | 1 fps | 74.92 | 75.53 | 74.10 | 73.08 | 74.44 | 59.92 | 76.14 | 62.91 | 62.16 | 45.80 | 67.63 |
| TimeChat-Online-7B (Yao et al., 2025) | | 1 fps | 80.22 | 82.03 | 79.50 | 83.33 | 76.10 | 78.50 | 78.70 | 64.63 | 69.60 | 57.98 | 75.36 |
| StreamBridge (Wang et al., 2026) | | 1 fps | 84.74 | 82.68 | 88.92 | 89.77 | 77.36 | 85.36 | 84.26 | 69.92 | 71.67 | 35.75 | 77.04 |
| **Open-source Offline VideoLLMs (+ Streaming-Adapted)** | | | | | | | | | | | | | |
| LLaVA-OneVision-7B | (uniform sampling) | 32 | 80.38 | 74.22 | 76.03 | 80.72 | 72.67 | 71.65 | 67.59 | 65.45 | 65.72 | 45.08 | 71.12 |
| (Li et al., 2024a) | (sliding window) | 32 | 84.28 | 78.91 | 90.22 | 85.26 | 81.13 | 77.88 | 73.15 | 71.14 | 76.14 | 37.23 | 77.40 |
| | + Uniform cache | 1 fps | 84.82 | 81.25 | 89.27 | 83.65 | 76.10 | 78.50 | 76.85 | 72.76 | 73.30 | 36.70 | 76.92 |
| | + DSCache | 1 fps | 86.72 | 78.12 | 86.12 | 89.00 | 79.62 | 84.42 | 77.78 | 74.39 | 77.84 | 36.70 | **79.12** |
| Qwen2.5-VL-7B | (uniform sampling) | 32 | 78.32 | 80.47 | 78.86 | 80.45 | 76.73 | 78.50 | 79.63 | 63.41 | 66.19 | 53.19 | 73.68 |
| (Bai et al., 2025) | (sliding window) | 32 | 81.30 | 81.10 | 88.01 | 86.17 | 81.01 | 83.18 | 81.48 | 70.73 | 71.02 | 43.09 | 77.61 |
| | + Uniform cache | 1 fps | 87.53 | 81.25 | 88.33 | 85.26 | 77.99 | 86.29 | 81.48 | 68.29 | 70.74 | 45.21 | 78.56 |
| | + DSCache | 1 fps | 88.62 | 79.69 | 90.85 | 89.00 | 78.98 | 92.52 | 82.41 | 79.67 | 79.55 | 40.43 | **82.32** |

comprising 644 videos with 3035 QA-pairs. RVS-Ego and RVS-Movie (Zhang et al., 2024) serve as additional streaming testbeds for long videos up to one hour in duration, featuring timestamped open-ended questions on 10 Ego4D videos (Grauman et al., 2022) and 22 MovieNet videos (Huang et al., 2020).

**Evaluation.** StreamingBench and OVO-Bench consist of multiple-choice questions and are evaluated using accuracy. RVS-Ego and RVS-Movie contain open-ended questions; model responses are assessed by an LLM for correctness (accuracy) and overall answer quality (score).

**Implementation Details.** We apply our method to two types of state-of-the-art MLLMs for long-video understanding; LLaVA-OV (Li et al., 2024a) and Qwen-2.5-VL (Bai et al., 2025). For feature encoding in the feature buffer, we adopt the default vision tower of each MLLM, specifically SigLIP for Llava-OV and Qwen2.5-VL's own pretrained Vision Transformer(ViT). Following prior work (Yao et al., 2025; Di et al., 2025), videos are processed at 1 FPS for StreamingBench and OVO-Bench, and 0.5 FPS for RVS-Ego and RVS-Movie. By default, the context window length is $l_W = l_I + l_U = 32$ frames. Within this window, DSCache maintains a feature buffer of $l_I = 4$ frames, and a cumulative KV cache of $l_U = 28$ frames, which is fully utilized, i.e. $l_L = l_U$. The streaming baseline with standard uniform KV caches corresponds to a special case of

DSCache with $l_I = 0$, and $l_U = l_W$. The total token budget is bounded by the context window length. For LLaVA-OV-7B, each frame produces 196 tokens (∼6.3K max), and for Qwen2.5-VL (448×448), 286 tokens (∼9.2K max). This excludes system prompt and query tokens, which add a small overhead. Additional implementation details (e.g. resolution scaling and temporal subsampling) are in Supplementary.

### 5.2. Main Results

Results on StreamingBench and OVO-Bench are reported in Table 1 and Table 2, where DSCache achieves state-of-the-art performance. By adapting offline models to streaming inference, DSCache consistently outperforms their offline counterparts with uniform sampling by an average of 7.5%, and surpasses the state-of-the-art online VideoLLM, StreamBridge and Timechat-online, by 5.0% on StreamingBench and 10.8% on OVO-Bench, respectively. The streaming also outperforms prior methods on OVO-Bench. Compared with this baseline, DSCache yields a further improvement of 2.5%, demonstrating the effectiveness of enhancing informativeness of encoded recent inputs. In particular, DSCache achieves substantial gains on subtasks that depend on rencency information. For instance, on StreamingBench, it improves attribute perception (ATP), text-rich understanding (TR), spatial understanding (SU), and action perception (ACP) by 4.5%, 6%, 6.5%, and 6.7%, respectively. These gains are accompanied by modest trade-offs on

*Table 2.* Comparisons on OVO-Bench, evaluated by accuracy. Uniform cache denotes uniform KV caching baseline.

| Model | | #Frames | Real-Time Percep. | Backward Tracing | Forward Resp. | Overall |
|---|---|---|---|---|---|---|
| **Proprietary MLLMs** | | | | | | |
| Gemini 1.5 pro (Team et al., 2024) | | 1 fps | 70.8 | 62.3 | 57.2 | 65.3 |
| GPT-4o | | 64 | 63.6 | 58.7 | 53.4 | 58.6 |
| **Open-source Online VideoLLMs** | | | | | | |
| Flash-VStream-7B (Zhang et al., 2024) | | 1 fps | 29.9 | 25.4 | 44.2 | 33.2 |
| VideoLLM-online-8B (Chen et al., 2024a) | | 2 fps | 20.8 | 17.7 | - | - |
| TimeChat-Online-7B (Yao et al., 2025) | | 1 fps | 61.9 | 41.7 | 36.7 | 46.7 |
| **Open-source Offline VideoLLMs (+ Streaming-Adapted)** | | | | | | |
| LLaVA-OneVision-7B | (uniform sampling) | 32 | 63.8 | 43.1 | 51.7 | 52.9 |
| | (sliding window) | 32 | 70.9 | 44.1 | 49.8 | 54.9 |
| | + Uniform cache | 1 fps | 68.4 | 46.0 | 52.9 | 55.6 |
| | + DSCache | 1 fps | 71.5 | 47.8 | 53.1 | **57.5** |
| Qwen2.5-VL-7B | (uniform sampling) | 32 | 59.6 | 39.4 | 40.4 | 46.6 |
| | (sliding window) | 32 | 70.6 | 43.3 | 44.3 | 52.7 |
| | + Uniform cache | 1 fps | 67.8 | 42.3 | 46.6 | 52.2 |
| | + DSCache | 1 fps | 71.7 | 44.8 | 49.7 | **55.4** |

subtasks emphasizing long-range reasoning, such as causal reasoning (CR) and counting (CT). Notably, the improvement does not stem from using stronger offline models. State-of-the-art methods, such as Timechat-online, also use competitive pretrained models (e.g. Qwen2.5-VL-7B). We further provide detailed comparisons with existing cache optimization methods in Table 4, showing our strong performance with minimum memory usage and modest latency increase. The results show that compared to cache optimization, targeting construction yields more consistent gains.

We evaluate DSCache on RVS-Ego and RVS-Movie in Table 3 using LLaVA-OV variants. RVS-Ego emphasizes long-range recall, benefiting the streaming baseline with accumulated KV caches, while offline models are limited by sparse sampling. RVS-Movie focuses on recent content, favoring offline models that use dense raw features. Across both datasets, DSCache achieves consistently strong performance, demonstrating its effectiveness in combining fine-grained recency modeling with accumulated past. We also provide additional results on the captioning task to show its scalability in the presence of text caches (chat history); please refer to the Supplementary for further details.

*Table 3.* Comparisons on the RVS-Ego and RVS-Movie benchmarks. 'Acc.' indicates accuracy, and 'Score' is the open-ended answer rating by gpt-3.5-turbo on a scale of $1 \sim 5$.

| Method | #Frames | RVS-Ego | | RVS-Movie | |
|---|---|---|---|---|---|
| | | Acc. | Score | Acc. | Score |
| LLaVA-OV-7B | 32 | 57.5 | 3.94 | 48.7 | 3.47 |
| + Uniform cache | 0.5 fps | 58.9 | 3.93 | 48.1 | 3.46 |
| + DSCache | 0.5 fps | **59.5** | **3.97** | **49.4** | **3.50** |
| LLaVA-OV-0.5B | 32 | 53.6 | 3.82 | 41.1 | 3.36 |
| + Uniform cache | 0.5 fps | 53.2 | 3.78 | 41.2 | 3.34 |
| + DSCache | 0.5 fps | **54.9** | **3.89** | **41.9** | **3.38** |

### 5.3. Ablation Study

**Cumulative KV cache length $l_U$.** We study the impact of cumulative KV cache length $l_U$ by fixing the feature buffer length $l_I$, and varying the context length $l_W = l_I + L_U$. For DSCache , we fix $l_I = 4$ frames, while the uniform streaming baseline is equivalent to set $l_I = 0$. As shown in Figure 3a, under DSCache , OVO-Bench consistently benefits from increasing $l_U$, whereas performance on StreamingBench degrades when $l_U$ becomes large. For the streaming baseline, we observe a clear trade-off with respect to $l_U$: short caches lack sufficient context, while overly long caches accumulate redundant or noisy information, degrading performance. In contrast, DSCache decouples instant and cumulative cache construction, making it more robust to variations in $l_U$.

**Feature buffer size $l_I$.** The buffer size $l_I$ directly affects the informativeness of the instant caches. In Figure 3b, we fix $l_U = 28$ frames and analyze the impact of varying $l_I$. Small $l_I$ preserves higher fidelity but fails to capture sufficient temporal context. A moderate buffer size provides a better balance, effectively capturing informative recent dynamics while remaining efficient. Notably, OVO-Bench shows larger gains with smaller $l_I$, indicating a stronger dependence on fine-grained recency context.

**Attention sinks $\mathcal{C}_a$.** Prior work highlights the importance of attention sinks, implemented as initial tokens. We examine their role in constructing cumulative and instant caches in Table 5. Removing sinks leads to a clear performance degradation, with a larger impact on instant cache construction, reflecting the strong dependence of these tasks to recent context. This behavior is expected, as models are trained with attention sinks, and removing them introduces a training–inference mismatch that adversely affects performance.

*Table 4.* Comparison with existing cache optimization methods. Runtime and memory are tested on NVIDIA H200 GPU with 1 FPS. StreamingVLM adopts uniform caching, equivalent to our uniform cache baseline.

| Method | Backbone | StreamingBench (Real-time) | OVO-Bench (Real / Backward / Forward) | Latency | Memory |
|---|---|---|---|---|---|
| DSCache | | 79.12 | 57.5 (71.5 / 47.8 / 53.1) | 0.30s | 16 GB |
| ReKV (Di et al., 2025) | LLaVA-OV-7B | 71.06 | 53.9 (62.0 / 48.5 / 51.2) | 0.23s | 21 GB |
| StreamingVLM (Xu et al., 2025) | | 76.92 | 55.6 (68.4 / 46.0 / 52.9) | 0.21s | 16 GB |
| DSCache | | 82.32 | 55.4 (71.7 / 44.8 / 49.7) | 0.49s | 17 GB |
| InfiniPot-V (Kim et al., 2026) | Qwen2.5-VL-7B | 76.40 | 53.6 (65.9 / 47.6 / 47.9) | 0.37s | 17 GB |
| StreamingVLM (Xu et al., 2025) | | 78.56 | 52.2 (67.8 / 42.3 / 46.6) | 0.31s | 17 GB |

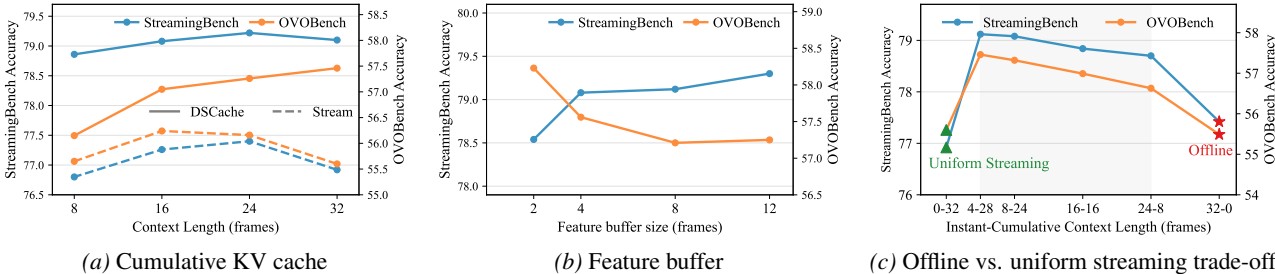

*(a)* Cumulative KV cache     *(b)* Feature buffer     *(c)* Offline vs. uniform streaming trade-off

*Figure 3.* Ablations on cumulative past KV cache and feature buffer for instant KV cache construction using LLava-OV-7B.

*Table 5.* Ablations on the effect of attention sinks in instant and cumulative cache construction.

| Base Model | Attention sink cum. | instant | StreamingBench | OVO-Bench |
|---|---|---|---|---|
| | ✓ | ✓ | **79.12** | **57.52** |
| Llava_OV_7B | ✗ | ✓ | 79.06 | 57.09 |
| | ✓ | ✗ | 74.27 | 55.20 |
| | ✓ | ✓ | **82.32** | **55.40** |
| Qwen2.5_VL_7B | ✗ | ✓ | 81.80 | 53.41 |
| | ✓ | ✗ | 76.31 | 50.15 |

**Sliding-window offline vs. uniform streaming trade-off.** Offline models with sliding-window sampling focus on recent context, whereas the streaming baseline with uniform KV caches accumulates long-range history. Given a context window of $l_U + l_I$, setting $l_I = 0$ reduces DSCache to the streaming baseline, while setting $l_U = 0$ degenerates it to a sliding-window offline model. In Figure 3c, we illustrate this trade-off by fixing the context window to 32 frames and varying $l_I$ and $l_U$. DSCache combines the strengths of both, preserving informative recent context while maintaining cumulative KV caches, leading to superior performance.

*Table 6.* Combining DSCache with pior methods (LLaVA-OV-7B).

| Method | StreamingBench | OVO-Bench (Real / Backward / Forward) |
|---|---|---|
| DSCache | 79.12 | 57.5 (71.5 / 47.8 / 53.1) |
| + ReKV | 79.41 | 58.5 (72.4 / 49.1 / 53.8) |
| + InfiniPot-V | 79.60 | 58.4 (71.8 / 48.9 / 54.7) |

**Compatibility with cache optimization methods.** Existing methods, e.g. ReKV (Di et al., 2025) and InfiniPot-V (Kim et al., 2026), focus on cache compression/retrieval

over long contexts. DSCache is complementary, prioritizing recency encoding to better capture current events. It can be combined with these methods to improve long-range modeling. We observe further gains when applying these to the cumulative past cache in Table 6.

### 5.4. Runtime Analysis

DSCache maintains the cumulative KV cache as in the streaming baseline but recomputes the instant cache on demand, trading computational efficiency for recent context informativeness. We evaluate runtime and memory usage on StreamingBench using an NVIDIA H200 GPU (Table 7). We report per-query latency, when a response is generated and latency is most relevant; the overhead for non-query frames is negligible for DSCache and comparable to the streaming baseline.

Offline models process 32 sampled frames at query time, while DSCache and the streaming baseline use the streaming implementation and maintain constant memory, enabling deployment on unbounded streams. Offline models must recompute all 32 sampled frames per query, resulting in high latency. For the streaming baseline and DSCache, latency is split into a prefilling stage for KV cache construction and a decoding stage for response generation. DSCache matches the streaming baseline in decoding latency but incurs additional prefilling cost to compute the instant cache. Nonetheless, the overall overhead remains much lower than offline inference and comparable to the streaming baseline, yielding a favorable performance–efficiency trade-off. Notably, this extra computation is triggered only when a user query

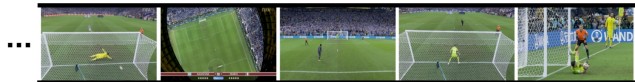

**What happens after the penalty kick is taken?**
   *A. The penalty kick was saved by the goalkeeper.* ⒟
   B. The goalkeeper catches the ball.
   *C. The ball goes into the bottom left of the net.* Ⓢ
   D. The ball is deflected by a defender.

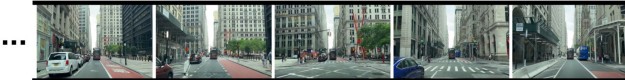

**What is written on the blue bus's side near the top?**
   A. City Lights Entertainment.
   *B. Starr.* ⒟
   *C. Broadway Tours.* Ⓢ
   D. Manhattan Transit.

*(a)* Successful cases.

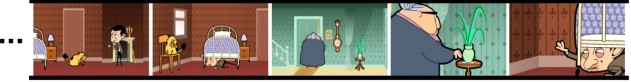

**Why is Mr. Bean under the bed now?**
   *A. He wanted to snatch back the teddy bear doll that was
      taken by the cat.* Ⓢ
   *B. He dropped something valuable and is trying to retrieve it.* ⒟
   C. He is hiding from someone who just entered the room.
   D. He thought he saw something strange under the bed.

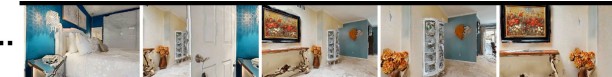

**Where are the orange flowers?**
   *A. In the living room, on the table next to the couch.* ⒟
   *B. Outside the bedroom, in a vase on the ground.* Ⓢ
   C. In the bathroom, on the counter near the sink.
   D. In the bedroom, on the nightstand next to the bed.

*(b)* Failure cases.

*Figure 4.* Qualitative analysis. Red shows incorrect predictions; green denotes the ground truth. Ⓢ: uniform streaming. ⒟: DSCache.

arrives and is not performed at every time step.

Admittedly, recomputing the instant cache in DSCache adds latency, which increases with respect to buffer size. However, latency remains within the input temporal resolution (1 FPS), making it suitable for streaming. We further propose an approximate variant that avoids full recomputation at each query, achieving computational complexity comparable to the uniform cache with only a minor performance trade-off. See more details in Supplementary.

*Table 7.* Runtime analysis on StreamingBench dataset using LLava-OV-7B with an NVIDIA H200 GPU.

| Model | GPU Memory | Latency(s) | | |
|---|---|---|---|---|
| | | Prefilling | Decoding | Overall |
| Offline | 19 GB | - | - | 2.85 |
| Uniform | 16 GB | 0.07 | 0.14 | 0.21 |
| DSCache | 16 GB | 0.16 | 0.14 | 0.30 |

### 5.5. Qualitative Results

Figure 4 shows qualitative results on StreamingBench. Due to the cumulative nature of uniform KV caching, the streaming baseline either attends to outdated context (left) or misses fine-grained visual details (right), highlighting the importance of preserving recent context, as achieved by DSCache. We further analyze attention distributions over 50 cases where DSCache is correct but the uniform baseline is not. Splitting the cache into recent (buffer) and past, DSCache allocates 67% of attention to recent, compared to 45% for the baseline, indicating stronger recency focus and better capture of current events in streaming scenarios.

However, cases requiring extensive long-range reasoning remain challenging for DSCache. Failure cases of

DSCache are shown in Figure 4, where required information lies in distant context or demands comprehensive long-range reasoning. These results highlight the inherent trade-off between capturing detailed recent information and retaining long-range dependencies.

## 6. Conclusion

This paper presents DSCache, a training-free method for adapting offline models to streaming inference under strict resource and efficiency constraints. Building on a position-agnostic encoding strategy, DSCache enables unbounded streaming by mitigating the issue of position extrapolation. Meanwhile, by decoupling cumulative and instant cache construction, it prevents the cumulative past KV cache from interfering with the instant cache construction, preserving informative recent context while enabling leverage the cumulative past KV cache. DSCache achieves superior performance on streaming video understanding tasks, especially those heavily rely on recent context. Future efforts will focus on improving the efficiency of the instant cache construction and augmenting DSCache with better cache optimization strategies to further enhance cumulative past KV cache construction and usage.

## Acknowledgment

This research/project is supported by the Ministry of Education, Singapore, under its MOE Academic Research Fund Tier 2 (MOE-T2EP20125-0037). We would like to acknowledge that computational work involved in this research is partially supported by NUS IT's Research Computing group using grant number NUSREC-HPC-00001.

## Impact Statement

This paper presents work aimed at improving streaming video understanding and advancing real-time inference capabilities. We do not identify specific societal consequences arising from this work.

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

## A. Cumulative Effect in Uniform Streaming Cache

Figure 5 compares three KV cache constructions for a newly arriving frame: 1) single frame, 2) offline (recomputed from sampled frames), and 3) streaming (uniform KV cache). Using the single-frame construction as reference, we evaluate how faithfully the latter two encode the frame, measuring cache similarity and the resulting performance. As shown in the figure, the streaming cache deviates more from this reference, indicating a cumulative effect from residual past information that weakens new caches. This effect is further reflected in the performance drop observed with the uniform cache baseline. To further quantify this, we fix the cache size for inference and vary how the current frame cache is constructed. As in Table 8, increasing context length in streaming cumulative cache increases residual accumulation, which reduces cache similarity and degrades performance. This indicates a strong correlation between cache similarity and task performance.

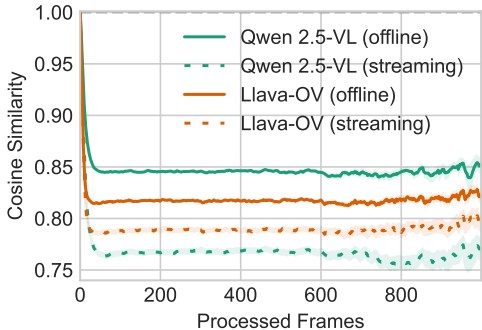

*Figure 5.* Value-cache similarity for the arriving frame, using single-frame cache as the reference. Compared to offline caches, uniform streaming caches accumulate past context, resulting in lower similarity, indicating that newly arriving frames are captured less accurately (the context window length is set to 32 frames).

*Table 8.* Similarity vs. performance on StreamingBench (LLaVA-OV-7B).

| Setting | Context | Cosine Similarity | Accuracy |
|---|---|---|---|
| Offline | | 0.82 | 77.96 |
| Streaming | 4s | 0.81 | 77.76 |
| Streaming | 8s | 0.79 | 77.20 |
| Streaming | 16s | 0.74 | 76.79 |
| Streaming | 32s | 0.71 | 76.67 |

## B. Proof

**Proposition 3.2.** For RoPE-based LLMs, using position-agnostic encoding with shifted position IDs that preserve relative distances between tokens produces identical outputs to standard positional encoding with absolute position IDs.

*Proof.* We compare two KV cache encoding strategies for RoPE-based LLMs with different treatment of positional information in attention calculation.

- **Standard positional encoding**: keys incorporate positional information at construction using absolute position IDs.

- **Position-agnostic encoding**: keys are stored before applying position transformation, and position IDs are reassigned starting from 0 when used, preserving relative distances.

Values are position-agnostic and thus identical under both strategies. For simplicity, we assume contiguous position IDs. The proof extends to non-contiguous cases as long as the same relative distances are preserved across these two strategies. As positional information affects the model only via self-attention, while subsequent residual connections and feed-forward networks are position-agnostic, it suffices to show that self-attention outputs are identical.

We prove equivalence by induction over layers and time. 1) Layer-wise: the first transformer layer receives identical inputs under both strategies and produces identical outputs; these outputs form the inputs to the next layer. By induction over

layers, the outputs of each corresponding layer are identical across the two strategies. 2) Time-wise: the same reasoning applies at past or future steps, implying equivalence for all time steps.

**Eviction-free.** Let the current input be $X_u = \{x_{i+l_u}\}_{i=0}^{n_u-1}$ with $x_i \in \mathbb{R}^d$ being the token embedding, and the past KV cache constructed from previous inputs $\{X_j\}_{j=0}^{u-1}$, where $l_u$ denotes the total number of observed tokens up to $u$, and $n_u$ denotes the length of $X_u$. Let $r$ index the transformer layers, Denote the standard KV cache as $\mathcal{P} = \{(K^{(r)}, V^{(r)})\}_{r=0}^{N-1}$, and the position-agnostic cache as $\mathcal{P}' = \{(K'^{(r)}, V^{(r)})\}_{r=0}^{N-1}$. Denote $Z^{(r)}$ as the input of layer $r$, where both strategies share the same input, with $Z^{(0)} = X$.

We start from the first transformer layer with $z_i^{(0)} = x_i$, and omit the layer index $r$ for notational simplicity. Values are updated by concatenate the past cache with the new one, as

$$V = [V, \{W_v z_i\}_{i=l_u}^{l_u+n_u-1}] = \{W_v z_i\}_{i=0}^{l_u+n_u-1},$$

and keys are updated as

$$K = [K, \{f(W_k z_i, i)\}_{i=l_u}^{l_u+n_u-1}] = \{f(W_k z_i, i)\}_{i=0}^{l_u+n_u-1},$$

$$K' = [K', \{W_k z_i\}_{i=l_u}^{l_u+n_u-1}] = \{W_k z_i\}_{i=0}^{l_u+n_u-1},$$

where the function $f$ incorporates position information, $W$ is a linear projection to hidden dimension $D$, and $[,]$ denotes concatenation operation along the sequence-length dimension. Since no cache eviction occurs, the reassigned position IDs in the position-agnostic encoding coincide with the absolute position IDs. These two forms can be aligned by applying the positional transformation to the position-agnostic keys:

$$K_i = f(K'_i, i)$$

Given the current queries with positional information encoded, $Q = \{f(W_q z_i, i)\}_{i=l_u}^{l_u+n_u-1}$, the attention weights computed under the two strategies are identical:

$$a_{ij} = softmax\left(\frac{Q_i K_j}{\sqrt{D}}\right) = softmax\left(\frac{f(W_q z_i, i)f(W_k z_j, j))}{\sqrt{D}}\right) = softmax\left(\frac{Q_i f(K'_j, j))}{\sqrt{D}}\right) = a'_{ij},$$

and the self-attention outputs are the same,

$$\sum_{j=0}^{l_u+n_u-1} a'_{ij} V_j = \sum_{j=0}^{l_u+n_u-1} a_{ij} V_j.$$

Applying position-agnostic residual and FFN operations, both strategies produce identical layer-wise outputs $\{z_i^{(1)}\}_{i=l_u}^{l_u+n_u-1}$, which serve as inputs to the next layer. By induction over layers, the outputs of each corresponding layer are identical across the two strategies. Similarly, by induction over time, this layer-wise equivalence extends to previous and future inputs.

**Case eviction.** Suppose at time step $t$, the past KV cache is truncated to a window of length $l_W$ to meet the resource constraints. Let $X_t = \{x_{i+l_t}\}_{i=0}^{n_t-1}$ be the current input. Given the total number of observed tokens $l_t$, the KV caches under both strategies can be represented

$$\mathcal{P}^{(r)} = \{K_i^{(r)}, V_i^{(r)}\}_{i=l_t-l_W}^{l_t-1}, \quad \mathcal{P}'^{(r)} = \{K_i'^{(r)}, V_i^{(r)}\}_{i=l_t-l_W}^{l_t-1}$$

Likewise, we first analyze the attention computation in the first transformer layer, and omit the layer index $r$ for simplicity.

- Standard KV cache. Given the new input sequence $\{x_i\}_{i=l_t}^{l_t+n_t-1}$, queries are computed using consecutive absolute position IDs starting from $l_t$,

$$Q = \{Q_i\}_{i=l_t}^{l_t+n_t-1} = \{f(W_q x_i, i)\}_{i=l_t}^{l_t+n_t-1}.$$

  Keys and values are updated by concatenating the cached ones with those computed from the new input:

$$K = \{K_i\}_{i=l_t-l_W}^{l_t+n_t-1} = [\{f(W_k x_i, i)\}_{i=l_t-l_W}^{l_t-1}, \{f(W_k x_i, i)\}_{i=l_t}^{l_t+n_t-1}] = \{f(W_k, x_i, i)\}_{i=l_t-l_W}^{l_t+n_t-1}$$

$$V = \{V_i\}_{i=l_t-l_W}^{l_t+n_t-1} = [\{W_v x_i\}_{i=l_t-l_W}^{l_t-1}, \{W_v x_i\}_{i=l_t}^{l_t+n_t-1}] = \{W_v x_i\}_{i=l_t-l_W}^{l_t+n_t-1}$$

$\forall i \in [l_t, l_t + n_t - 1], j \in [l_t - l_W, l_t + n_t - 1]$, the attention weight between the query $Q_i$ and key $K_j$ is computed as

$$a_{ij} = softmax(\frac{Q_i K_j}{\sqrt{D}}),$$

with

$$Q_i K_j = <f(W_q x_i, i), f(W_k x_j, j)> = g(W_q x_i, W_k x_j, i - j)$$

where the last equality to the relative form $g$ follows from the property of RoPE that the inner product depends only on the relative position $i - j$.

- Position-agnostic KV cache. Since value caches do not encode positional information, given the new input sequence, values are updated identically to the standard strategy,

$$V = \{W_v x_i\}_{i=l_t-l_W}^{l_t+n_t-1}$$

Keys are first updated to incorporate new input, after which position information is incorporated using consecutive position IDs ranging from 0 to $l_W + n_t - 1$, namely by subtracting the offset $l_t - l_W$

$$K' = \{W_k x_i\}_{i=l_t-l_W}^{l_t+n_t-1}$$
$$K'' = \{f(W_k x_i, i - l_t + l_W)\}_{i=l_t-l_W}^{l_W+n_t-1}$$

where $K'$ and $V$ will be stored as the new KV cache.

Queries are computed using position IDs consistent with those of the newly added keys, i.e. from $l_W$ to $l_W + n_t - 1$.

$$Q' = \{f(W_q x_i, i - l_t + l_W)\}_{i=l_t}^{l_t+n_t-1}.$$

$\forall i \in [l_t, l_t + n_t - 1], j \in [l_t - l_W, l_t + n_t - 1]$, the inner product between updated keys and queries becomes

$$Q_i' K_j'' = <f(W_q x_i, i - l_t + l_W), f(W_k x_j, j - l_t + l_W)> = g(W_q x_i, W_k x_j, i - j) = Q_i K_j$$

Therefore, the attention weights satisfy

$$a_{ij}' = softmax\left(\frac{Q_i' K_j''}{\sqrt{D}}\right) = softmax\left(\frac{Q_i K_j}{\sqrt{D}}\right) = a_{ij},$$

As shown above, because both the attention weights and value vectors are identical under the two strategies, their self-attention outputs are identical. By induction over layers and time, the outputs of each corresponding layer are identical across the two strategies.

$\square$

## C. Implementation

For StreamingBench and OVO-Bench, videos are processed at 1 FPS. The context window length $l_W = l_I + l_U$ for these datasets is set to 32 frames. DSCache maintains a feature buffer of $l_I = 4$ frames for recent inputs, while the cumulative past KV cache covers the remainder of the context window and is fully utilized, i.e. $l_L = l_U$. In addition, we perform an extra operation on the feature buffer by doubling the spatial resolution of the very recent frame to capture more fine-grained details. In the Qwen-2.5-VL implementation, which generates more tokens per frame, we also sample the cumulative KV cache at 0.5 FPS during decoding to reduce redundancy over the long-range context.

For RVS-Ego and RVS-Movie, we use a sampling rate of 0.5 FPS. As queries in these two datasets involve long context recall, we adopt a longer context window of $l_W = 256$ frames, while updating the cumulative KV cache using only the most recent 32 frames, i.e. $l_L = 32$ frames. The feature buffer is set to $l_I = 4$ frames for RVS-Ego, and $l_I = 8$ frames for RVS-Movie. No additional operations such as increased spatial resolution or cache sampling are applied.

Since all of the above streaming VQA datasets involve independent queries, the cumulative KV cache stores only visual token representations and discards the text tokens corresponding to each query.

# D. Experimental Results

## D.1. Captioning

We evaluate DSCache on an extra caption task using LiveCC3K-Sports-CC (Chen et al., 2025), a dataset of real-time sports commentary that captures rich spatiotemporal semantics. Models are prompted with the video title and preceding CCs to generate live commentary. Predictions from two models are compared pairwise, with GPT-4o selecting the better one based on the ground-truth CCs, and winning rates are used to rank the models. We adapt StreamingVLM (Xu et al., 2025) by replacing its uniform streaming cache with DSCache. Although StreamingVLM is pretrained and deployed with a small context window of 16 frames, which partially mitigates the cumulative effect in the streaming cache, incorporating DSCache in a training-free manner achieves an average winning rate of 50.8±0.2% over 3 runs, indicating a modest but consistent improvement in real-time captioning.

## D.2. Extra Ablation

**Cache decoupling.** Our design decouples the feature buffer from the cumulative past KV cache, delaying the caching of recent features. The uniform cache baseline serves as the non-decoupled case, and the effect of cache decoupling is reflected in the performance comparison with the baseline where consistent gains in DSCache show its effectiveness.

**Position-agnostic encoding.** It enables flexible cache operations in unbounded streams. Standard position-aware caching is incompatible with cache eviction, causing discontinuous positions; in long streams, it leads to out-of-distribution positions and overflow. We test position-aware caching in the presence of discontinuity and out-of-distribution positions on StreamingBench with LLaVA-OV-7B. Performance drops from 79.12 to 77.84, highlighting the need of our encoding. Beyond the proof, we empirically validate their equivalence. In a controlled setting without eviction and positional overflow (both encodings are applicable), we evaluate a subset of StreamingBench ($\leq$ 5 minutes, 32 videos with 160 queries, LLaVA-OV-7B, 1 FPS). Results show identical performance (79.38%), confirming equivalence.

**Spatial resolution** In contrast to uniform streaming caches, which must operate at a fixed resolution throughout, the decoupling enables additional processing on features of recent inputs, such as increasing the spatial resolution of recent frames, without affecting the cumulative past cache construction and update efficiency. In Table 9, we study the impact of increasing the resolution of the latest frame and show that incorporating finer visual details, even for a single frame, can yield further performance gains. Notably, we observe consistent performance improvements even without any resolution increase, confirming that the gains arise from decoupling the construction of the cumulative and instant caches.

*Table 9.* Ablations on spatial resolution.

| Base Model | Method | Spatial | StreamingBench | OVOBench |
|---|---|---|---|---|
| **Llava-OV-7B** | Uniform | - | 76.92 | 78.56 |
| | DSCache | - | 78.88 | 81.32 |
| | | × 1.5 | 78.98 | 82.03 |
| | | × 2.0 | 79.12 | 82.32 |
| **Qwen2.5-VL-7B** | Uniform | - | 55.60 | 52.21 |
| | DSCache | - | 56.68 | 54.33 |
| | | × 1.5 | 57.02 | 54.95 |
| | | × 2.0 | 57.52 | 55.40 |

# E. Latency Overhead and Approximate Inference

In runtime analysis, we report latency at query time, i.e. per-query latency, when a response is generated and latency is most relevant; the overhead for non-query frames is negligible and comparable to the uniform cache baseline. Besides, we adopt cache eviction to maintain a fixed-size cache, bounded by a predefined context window (buffer size + cumulative past cache); so per-query latency and memory usage remains constant and does not scale with input length.

Latency mainly comes from instant cache construction at query frames. Table 10 shows larger buffers increase latency. In our setting, a small buffer (e.g. 4 frames) achieves high performance over uniform cache (79.12 vs. 76.92) while keeping latency manageable (0.30s vs. 0.21s).

To improve efficiency, we further propose an approximate design that avoids fully recomputing the instant cache at each

*Table 10.* Buffer size ablation (LLaVA-OV-7B, StreamingBench).

| Buffer Size (#frames) | Latency | Accuracy |
|:---:|:---:|:---:|
| 2 | 0.26 | 78.74 |
| 4 | 0.30 | 79.12 |
| 8 | 0.35 | 79.08 |
| 12 | 0.39 | 79.24 |

*Table 11.* Instant cache approximation (LLaVA-OV-7B, StreamingBench, 8-frame buffer). Larger periods lead to greater performance degradation due to residual accumulation.

| Method | Period (s) | Latency (s) | Accuracy |
|:---|:---:|:---:|:---:|
| Uniform Caching | – | 0.21 | 76.92 |
| DSCache | – | 0.35 | 79.08 |
| DSCache (approx.) | 4 | 0.22 | 78.86 |
|  | 8 | 0.23 | 78.98 |
|  | 16 | 0.22 | 78.48 |

query. It maintains an extra periodically refreshed cache for recent frames, enabling reuse of buffer-frame caches while preventing residual accumulation. Table 11 shows that this largely reduces latency, bringing it close to the uniform baseline. Latency also remains stable as buffer size increases(0.22s up to a 16-frame buffer). However, this approximation introduces a mild trade-off: periodic resets vary context across frames, slightly affecting performance.

## F. Qualitative Results

Figure 6 presents additional qualitative results on OVO-Bench, illustrating the benefits of maintaining informative recent context. The streaming baseline is distracted by earlier context and produces hallucinated outputs due to the uniform cumulative KV caching.

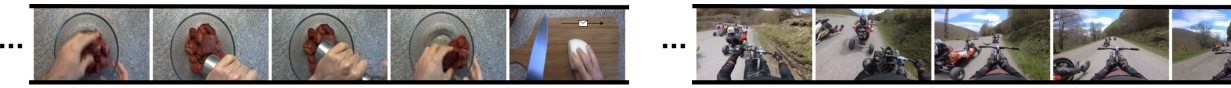

**What does the person do before cut the onion into slices?**
 *A. Add water to the pot.*
 *B. Mix paprika and pepper with the meat.* Ⓓ
 *C. Add oil to a pot.* Ⓢ
 *D. Cook the onions in the pot.*

**How many go-karts are visible in the area?**
 *A. One go-kart.*
 *B. Two go-karts.*
 *C. Three go-karts.* Ⓢ
 *D. Four go-karts.* Ⓓ

*Figure 6.* Successful cases on OVO-Bench. Text in red denotes the false predictions, while text in green represents the ground truth. Ⓢ - Streaming baseline. Ⓓ - DSCache.

