# OpenReview forum: "Decouple and Cache: KV Cache Construction for Streaming Video Understanding"
_ICML.cc/2026/Conference — ICML 2026 regular_

### Official Review · Reviewer_ruHv · 2026-03-08

**Soundness:** 3
**Presentation:** 2
**Significance:** 1
**Originality:** 2
**Overall Recommendation:** 4
**Confidence:** 3

**Summary:**

This paper introduces DSCache, which is a training-free KV cache compression algorithm aiming to reduce tokens from input videos; in an online streaming video setting, the number of KV cache from frames increase consistently, hence requiring such techniques.
In a high level, DSCache has two separated streams: cumulative past KV cache for long context, and recent feature buffer to preserve fine-grained details. Combining these two with additional tricks such as attention sink and handing RoPE, DSCache shows noticeable improvements in the performance.

**Compliance With Llm Reviewing Policy:**

Affirmed.

**Final Justification:**

I still find the contributions somewhat incremental. However, as all fellow reviewers recommends acceptance, and I am not strongly opposed to rejection, I am raising my score.

**Key Questions For Authors:**

What's the exact number of input tokens and FLOPs in Table 1, 2? Is it indeed using more computation than Uniform cache, and if so, how much?

**Limitations:**

The societal impact is briefly mentioned in the end of the paper. However, the limitations of this paper is not specified.

**Strengths And Weaknesses:**

The method is clear, and easy to understand.

The soundness is good. The idea of having coarser- & fine- grained streams have been widely used in the vision area, and DSCache makes use of it well.

The overall presentation is fair, but I do find that it's not that straightforward. For example, I believe the proof section in the supplementary is unnecessary as they are somewhat obvious. Instead, more details of the method (e.g., which feature the feature buffer uses to retain local details) are necessary. Also, how much tokens is it actually preserving in Table 1 and 2? The number of input tokens directly affect the FLOPs count, so the authors must list the exact token counts, or FLOPs in the table.

The significance & originality are both somewhere in fair-to-poor. I believe the method is very technical, and the novelty is limited. For instance, if one is to build a video LLM, having correct positional embeddings is not a novelty, but rather an obvious to-do.

---

> ### Author Rebuttal · Authors · 2026-03-31
>
> Dear reviewer ruHv
>
> Thank you for your time and constructive feedback. We are glad that the clarity and soundness of our method, along with its coarse- and fine-grained design, are appreciated. We address your questions below.
>
> **▶ W1. Presentation**
>
> We appreciate the clarity suggestions to improve our submission.
> - The proof provides a formal justification of our design, addressing a gap where prior works rely on heuristics and improves rigor. While the core intuition is straightforward, we include it for completeness and will condense it for readability.
> - We will clarify implementation details. Specifically, the feature buffer stores visual tokens extracted from the vision encoder (e.g., SigLIP / ViT) after LLM projection. In addition, we will explicitly report the token budget used in Tab. 1 and 2 in revision.
>
> **▶ W2. Novelty**
>
> To clarify, DSCache is not a KV cache compression method. Instead, it mainly studies how cache construction in streaming inference, i.e., what information to retain in the cache, affects representation quality.
> - We are the first to identify that cumulative caching in streaming settings introduces residual interference from past caches, even after eviction; this interference degrades the representation of newly incoming inputs. This phenomenon is largely underexplored, as recognized by reviewers nd5C, QsWm, and hugu.
> - We introduce position-agnostic encoding that decouples cache representations from positions to support unbounded long stream inference. This encoding also enables flexible cache operations, including eviction, partitioning, and recomposition via post-hoc position reassignment. In contrast, although using correct absolute positions in position-aware encoding is straightforward, it can suffer from out-of-distribution and overflow in long streams. Cache eviction also introduces discontinuous position indices, leading to a performance drop (from 79.12% to 77.84% on StreamingBench), which limits scalability to unbounded streams.
> - DSCache is complementary to existing cache utilization techniques such as compression and retrieval. It can be combined with these methods to further improve long-range modeling. We observe additional gains when applying them to our cumulative past KV cache.
>
>
> *Tab A. Combining DSCache with existing methods(LLaVA-OV-7B)*
> | Method | StreamingBench | OVOBench (Real / Backward / Forward) |
> |-------|-------|----------|
> | DSCache | 79.12 | 57.5 (71.5 / 47.8 / 53.1) |
> | + ReKV [1] | 79.41 | 58.5 (72.4 / 49.1 / 53.8) |
> | + InfiniPot-V [2] | 79.60 | 58.4 (71.8 / 48.9 / 54.7) |
>
> [1] Streaming Video Question-Answering with In-context Video KV-Cache Retrieval, ICLR, 2025
>
> [2] Memory-Constrained KV Cache Compression for Streaming Video Understanding, NeurIPS, 2025
>
> **▶ Q1. Experimental details**
>
> We provide the requested implementation details below.
> - We adopt the default vision tower in MLLMs for feature encoding, which is SigLIP for Llava-OV and Qwen2.5-VL’s own pretrained Vision Transformer(ViT).
> - In Tab. 1 and 2, the token budget is bounded by the context window: a buffer of 4 frames and 28 past frames for cumulative caching (32 frames total). For LLaVA-OV, each frame produces 196 tokens (\~6.3K max), and for Qwen2.5-VL (448×448), 286 tokens (\~9.2K max). This excludes system prompt and query tokens, which add a small overhead.
> - For fair comparison, the uniform cache baseline also maintains caches from the most recent 32 frames, ensuring equal token budgets. Therefore, our method does not increase the token count, but instead improves performance by enhancing the quality of cached tokens through better construction.
>
>
> **▶ Q2. Limitations**
>
> Thank you for pointing this out. We will include a dedicated discussion of limitations in the revision. In particular, our method introduces some additional overhead due to cache reconstruction at query time, which we will clarify and discuss along with our proposed solutions for improvement in revision.

---

> > ### Author Rebuttal · Reviewer_ruHv · 2026-04-02
> >
> > I appreciate the authors' response. I still have multiple concerns about this paper
> >
> > - "We are the first to identify that cumulative caching in streaming settings introduces residual interference from past caches, even after eviction; this interference degrades the representation of newly incoming inputs. This phenomenon is largely underexplored, as recognized by reviewers nd5C, QsWm, and hugu."
> >
> > I think this statement is quite ambiguous, could you please articulate?
> >
> > - "We introduce position-agnostic encoding that decouples cache representations from positions to support unbounded long stream inference. This encoding also enables flexible cache operations, including eviction, partitioning, and recomposition via post-hoc position reassignment. In contrast, although using correct absolute positions in position-aware encoding is straightforward, it can suffer from out-of-distribution and overflow in long streams. Cache eviction also introduces discontinuous position indices, leading to a performance drop (from 79.12% to 77.84% on StreamingBench), which limits scalability to unbounded streams."
> >
> > The original motivation of RoPE is indeed to remove the position-variant property of absolute positional embeddings. It is very well known that RoPE is essentially similar to [A], which is why I believe it is very difficult to view this as a novelty.
> >
> > [A] Train Short, Test Long: Attention with Linear Biases Enables Input Length Extrapolation. Ofir Press, Noah A. Smith, Mike Lewis.
> >
> > - "In particular, our method introduces some additional overhead due to cache reconstruction at query time"
> >
> > I also noticed this limitation while reading the paper. What's the exact wall clock time of this method? Since KV cache are projected each query, I believe this is indeed not a negligible overhead. Rather, I suspect this would be slower than highly optimized methods that use even more KV cache.

---

> > > ### Author Response · Authors · 2026-04-03
> > >
> > > Dear reviewer ruHv
> > >
> > > Thank you for the comment. We will clarify these points below.
> > >
> > > **1. Cumulative effect**
> > >
> > > We clarify the “cumulative effect” with a concrete example. In standard streaming KV caching, each new cache is constructed by attending to existing past caches. Consider a sequence of inputs A, B, and C. The cache corresponding to B is computed conditioned on the cache of A, while C is computed conditioned on both A and B. As a result, the cache for C will implicitly contain information originating from A. When A is later evicted due to memory constraints, its influence is not fully removed, as it has already been embedded into the cache of B and C. Consequently, when a new input D arrives, its representation is computed based on B and C, which still carry residual information from A.
> > >
> > > This demonstrates that eviction removes entries but not their propagated influence. We refer to this phenomenon as the cumulative effect, where historical information is recursively carried forward and interferes with the encoding of new inputs. We believe this issue has been underexplored in prior streaming KV cache designs, which primarily focus on cache utilization (e.g., compression or retrieval) rather than the quality of cache construction itself.
> > >
> > > **2. Position-agnostic encoding**
> > >
> > > We agree that RoPE enables relative position awareness in attention. However, we would like to clarify that RoPE still takes absolute position indices as inputs to the positional transformation, rather than explicitly modeling relative positions. The relative property only arises during the interaction between queries and keys, where their relative positional differences influence the attention computation.
> > >
> > > Nevertheless, relying on absolute position indices as in standard RoPE  is ill-suited for streaming settings.
> > > - As the sequence grows unbounded, position indices can exceed the range observed during training, leading to out-of-distribution behavior and potential position overflow.
> > > - Cache eviction introduces discontinuities in position indices, breaking the continuity assumption used during training. Pretrained MLLMs are typically trained under continuous positional assignments without cache eviction. Such discontinuities can therefore degrade performance. For example, consider a sequence with positions [1,…,12], where positions 1\~4 correspond to the system prompt and 5\-12 to two frames (each consisting of 4 tokens). When a new frame comes and the older visual cache (positions 5\~8) is evicted, the new frame is assigned absolute positions 13\~16.  As a result, the remaining cache contains discontinuous position indices, i.e., 1\~4 and 9\~16, which breaks positional continuity.
> > >
> > > The position-agnostic encoding avoids these issues by decoupling KV cache storage from positional encoding, allowing positions to be reindexed as a contiguous range at usage time. This ensures that (i) position indices remain within a bounded, in-distribution range, and (ii) discontinuities introduced by cache eviction are eliminated. (iii) flexible position handling under cache mutation (e.g., eviction and recomposition). Moreover, we provide an inductive proof of the correctness of the position re-indexing, offering a principled justification beyond prior heuristic usage.
> > >
> > > **3. Wall time**
> > >
> > > We report per-query latency, when a response is generated and latency is most relevant (the overhead for non-query frames is negligible and comparable to the uniform cache baseline). Admittedly, recomputing the instant cache from the feature buffer does add latency, which also increases with respect to buffer size (Tab. A). In our setting, a small buffer (e.g., 4 frames) achieves high performance over uniform cache (79.12 vs. 76.92) while keeping latency manageable (0.30s vs. 0.21s in Tab. 5 in main paper). Meanwhile, latency remains within the input temporal resolution (1 FPS), making it suitable for streaming.
> > >
> > > *Tab. A Buffer size(#frames) ablation (LLaVA-OV-7B, StreamingBench)*
> > > |Buffer Size|Latency|Acc.|
> > > |----|---|-----|
> > > |2 |0.26|78.74|
> > > |4 |0.30|79.12|
> > > |8 |0.35|79.08|
> > > |12|0.39|79.24|
> > >
> > > To further improve efficiency, we propose an approximation that avoids fully recomputation at each query. It maintains a periodically refreshed cache for recent frames, enabling reuse of buffer-frame caches while preventing residual accumulation. This reduces latency (Tab. B), bringing it close to the uniform baseline. Latency also remains stable as buffer size increases (~0.22s up to a 16-frame buffer).  However, this approximation introduces a mild trade-off: periodic resets vary context across frames, slightly affecting performance.
> > >
> > > *Tab. B. Instant cache approx. (Llava-ov-7B, StreamingBench, a 8-frame buffer). A large period hurts performance due to residual accumulation.*
> > > |Method|Period (s)|Latency (s)|Acc.|
> > > |----|----|----|---|
> > > |Uniform Caching|-|0.21|76.92|
> > > |DSCache|-|0.35|79.08|
> > > |DSCache with approximation |m=4|0.22|78.86|
> > > | |m=8|0.23|78.98|
> > > | |m=16|0.22|78.48|

---

### Official Review · Reviewer_huqu · 2026-03-11

**Soundness:** 3
**Presentation:** 3
**Significance:** 2
**Originality:** 3
**Overall Recommendation:** 4
**Confidence:** 4

**Summary:**

The paper addresses the core challenge of unbounded video stream processing in streaming video understanding and proposes a training-free Decoupled Streaming Cache mechanism (DSCache). Its core goal is to adapt offline pre-trained multimodal models to unbounded streaming inference scenarios, solving the issues where the cumulative effect of existing uniform KV caching leads to the loss of recent input information, and where position encoding overflow fails to support position extrapolation beyond the training length. DSCache preserves fine-grained information of recent inputs by separating the historical KV cache and the immediate cache; simultaneously, it introduces a position-agnostic encoding strategy to avoid position overflow and support flexible cache operations. The paper verifies the effectiveness of the method on multiple streaming VQA benchmarks. Core contributions include revealing the cumulative effect of uniform KV caching, proposing the paradigm of decoupled cache construction, designing the position-agnostic encoding strategy with formal proof, and achieving training-free adaptation from offline models to unbounded streaming scenarios.

**Compliance With Llm Reviewing Policy:**

Affirmed.

**Final Justification:**

The rebuttal changed my evaluation. The supplementary experiments and analyses resolved my primary concerns. I am raising my overall recommendation from 2 to 4.

**Key Questions For Authors:**

1.	Regarding key similar methods in KV cache optimization such as ReKV and StreamingVLM, why were direct quantitative comparisons not conducted on the main experimental benchmarks under the same parameters and scenarios? Can you provide such comparisons and explain the core advantages and gaps of DSCache relative to these methods in terms of performance and efficiency?
2.	The paper explicitly claims that DSCache is orthogonal to existing KV cache utilization technologies. Why were no combination experiments designed, such as "DSCache + ReKV "? Can you supplement these experiments to verify the feasibility and performance gains of combining them, and quantify the value of DSCache as an upstream building module for improving existing methods?
3.	Why were ablation studies not conducted for the two core designs: "cache decoupling" and "position-agnostic encoding"? Can you supplement control experiments that retain only cache decoupling or only position-agnostic encoding to clarify their contributions and impacts?
4.	The analysis of qualitative results remains at the level of phenomenon description without exploring differences in model inference behavior. Can you supplement analyses such as model attention heatmaps, KV cache feature encoding similarity, and intermediate inference generation results, and provide a deep reasoning behavior decomposition for the failure cases of DSCache?


After answering the above questions, I will consider raising the soundness and presentation scores to 3.

**Limitations:**

No. The "Impact Statement" only mentions no specific societal consequences. It is recommended to supplement the analysis of the method's limitations regarding model generality and task adaptability.

**Strengths And Weaknesses:**

**Strengths:**

1.	The position-agnostic encoding strategy provides a formal equivalence proof for RoPE-based LLMs, ensuring output consistency with standard positional encoding, which demonstrates a solid theoretical foundation.
2.	It reveals the cumulative effect of uniform KV caching, an issue that has not received sufficient attention, providing a new research perspective for future cache design.



**Weaknesses:**

1.	Although related works like ReKV and StreamingVLM are explicitly mentioned as key similar methods in the field of KV cache optimization, there is no direct quantitative comparison in the main experiments. The actual effectiveness is questionable.
2.	The paper explicitly states that the method operates during the construction stage of the KV cache and is orthogonal to existing technologies. However, no experimental proof is provided, casting doubt on its feasibility and effectiveness.
3.	Figure 2 is chaotic in form, the process flow is obscure, and the logical chain appears disconnected, making it difficult to understand the method's workflow.
4.	There is a lack of ablation studies for the two major designs: "cache decoupling" and "position-agnostic encoding," missing the feasibility and effectiveness analysis of each component respectively.
5.	The analysis of visualized qualitative results is superficial, without exploring the underlying differences in model inference behavior.

---

> ### Author Rebuttal · Authors · 2026-03-31
>
> Dear reviewer huqu
>
> We sincerely thank you for your thorough and constructive feedback, and for recognizing our formal proof, cumulative effect identification, and cache design. We address your questions below and hope our clarifications improve your evaluation.
>
> **▶ W1/Q1. Comparison with cache optimization**
>
> Thanks for the point. We focus on improving cache construction, i.e., what to store, which is orthogonal to cache optimization that targets compression/retrieval under a fixed budget (see W2). Tab. A provides detailed comparisons, showing strong performance with modest latency. Compared to optimization, targeting construction yields more consistent gains.
>
> *Tab A. Comparison with existing methods(runtime tested on NVIDIA H200 GPU, StreamingBench, 1fps). StreamingVLM [3] adopts uniform caching, equivalent to our uniform cache baseline.*
> |Method|Backbone|StreamingBench Real-time|OVOBench (Real / Backward / Forward)|Latency|Memory|
> |--|--|--|--|--|--|
> |DSCache|LLaVA-OneVision-7B|79.12|57.5 (71.5 / 47.8 / 53.1)|0.30s|16 GB|
> |ReKV [1]|LLaVA-OneVision-7B|71.06|53.9 (62.0 / 48.5 / 51.2)|0.23s|21 GB|
> |StreamingVLM [3]|LLaVA-OneVision-7B|76.92|55.6 (68.4 / 46.0 / 52.9)|0.21s|16 GB|
> |DSCache|Qwen2.5-VL-7B|82.32|55.4 (71.7 / 44.8 / 49.7)|0.49s|17 GB|
> |InfiniPot-V [2]|Qwen2.5-VL-7B|76.40|53.6 (65.9 / 47.6 / 47.9)|0.37s|17 GB|
> |StreamingVLM [3]|Qwen2.5-VL-7B|78.56|52.2 (67.8 / 42.3 / 46.6)|0.31s|17 GB|
>
> [1] Streaming Video Question-Answering with In-context Video KV-Cache Retrieval, ICLR, 2025
>
> [2] Memory-Constrained KV Cache Compression for Streaming Video Understanding, NeurIPS, 2025
>
> [3] Real-Time Understanding for Infinite Video Streams, ICLR, 2026
>
> **▶ W2/Q2.  Compatibility with cache optimization**
>
> Thanks for the suggestion. Existing methods[1, 2] focus on compression/retrieval over long contexts. DSCache is complementary, prioritizing recency encoding to better capture current events. It can be combined with these methods to improve long-range modeling. We observe further gains when applying these to the cumulative past cache.
>
> *Tab B. Combining DSCache with existing methods(LLaVA-OV-7B)*
> |Method|StreamingBench|OVOBench (Real / Backward / Forward)|
> |--|--|--|
> |DSCache|79.12|57.5 (71.5 / 47.8 / 53.1)|
> |+ ReKV [1]|79.41|58.5 (72.4 / 49.1 / 53.8)|
> |+ InfiniPot-V [2]|79.60 |58.4 (71.8 / 48.9 / 54.7)|
>
> **▶ W3.  Unclarity in Fig. 2**
>
> Thanks. We will revise Fig. 2 to improve clarity.
>
> **▶ W4/Q3.  Ablation for cache decoupling and position-agnostic encoding**
>
> These components serve different roles. We will clarify in revision.
>
> The effectiveness of cache decoupling is reflected in consistent performance gains in Tab. 1, 2, where the uniform cache baseline serves as the non-decoupled case.
>
> Position-agnostic encoding enables flexible cache operations in unbounded streams. Standard position-aware caching is incompatible with cache eviction, causing discontinuous positions; in long streams, it leads to out-of-distribution positions and overflow.
> - We test position-aware caching in the presence of discontinuity and out-of-distribution positions. Performance drops from 79.12 to 77.84, highlighting the need of our encoding.
> - Beyond the proof, we empirically validate their equivalence. In a controlled setting without eviction and positional overflow (both encodings are applicable), we evaluate a subset of StreamingBench (<5 minutes, 32 videos with 160 queries, LLaVA-OV-7B, 1 FPS). Results show identical performance (79.38%), confirming equivalence.
>
> **▶ W5/Q4. Model behavior analysis**
>
> Thanks for the suggestion. We provide additional analyses below.
>
> Extending Fig. 5, we measure cosine similarity between caches and its effect on performance. We fix cache size for inference and modify the current frame cache construction (single-frame, with a feature buffer, or from cumulative cache). Using the single-frame as reference,  Tab. C shows that longer context in the cumulative cache increases residual accumulation, reducing similarity and degrading performance. This indicates a strong correlation between cache similarity and performance, supporting our instant cache design.
>
> *Tab C. Similarity vs. performance (StreamingBench, Llava-OV-7B)*
> |Setting| |Cos Similarity|Acc.|
> |--|--|--|--|
> |Buffer| |0.82|77.96|
> |Cumulative Cache|4s|0.81|77.76|
> |  |8s|0.79|77.20|
> |  |16s|0.74|76.79|
> |  |32s|0.71|76.67|
>
> We also analyze attention distributions over 50 cases where DSCache is correct but the uniform baseline is not. Splitting the cache into recent (buffer) and past, DSCache allocates ~67% of attention to recent, compared to ~45% for the baseline, indicating stronger recency focus and better capture of current events in streaming scenarios.
>
> For failure analysis, we observed a global–recency trade-off: the model improves recency modeling but underperforms the baseline on tasks requiring global reasoning(e.g., Causal Reasoning). Due to time and space limits, we will include additional visualizations in later revision.

---

> > ### Author Rebuttal · Reviewer_huqu · 2026-04-04
> >
> > My concerns have been adequately addressed. I will increase my score accordingly.

---

### Official Review · Reviewer_QsWm · 2026-03-12

**Soundness:** 3
**Presentation:** 3
**Significance:** 3
**Originality:** 3
**Overall Recommendation:** 4
**Confidence:** 5

**Summary:**

This paper addresses unbounded streaming inference for streaming video understanding and proposes DSCache (Decoupled Streaming Cache). The core observation is that in uniform streaming KV cache construction, new frame caches are "contaminated" by residual information from historical caches, degrading the encoding quality of recent frames. To address this, the authors decouple the historical KV cache (cumulative past cache) from the immediate cache of current frames, using a feature buffer to store raw features of recent frames. At inference time, the instant cache is constructed independently and then concatenated with the cumulative cache. Additionally, the authors introduce position-agnostic encoding, deferring positional information to usage time, with a formal equivalence proof under RoPE. Experiments on StreamingBench, OVO-Bench, and RVS-Ego/Movie show consistent improvements, averaging around 2.5% accuracy gains.

**Compliance With Llm Reviewing Policy:**

Affirmed.

**Final Justification:**

All my concerns are addressed, so I maintain my score.

**Key Questions For Authors:**

Please see weakness.

**Limitations:**

yes

**Strengths And Weaknesses:**

- Strength
The identification of the cumulative effect is among the most valuable contributions of this paper. Figure 5 quantifies the interference of uniform streaming caches on new frame encoding via cosine similarity, which is intuitive and compelling — this analytical angle has rarely been systematically examined in prior work.

 DSCache requires zero modification to model parameters at inference time. The feature buffer stores only raw features rather than KV caches, and the instant cache is constructed on demand only when a query arrives. The additional compute cost is modest (prefilling increases from 0.07s to 0.16s, overall latency from 0.21s to 0.30s per Table 5), making the design practical for deployment.

Prior works such as StreamingLLM only provided heuristic justifications for position re-indexing. This paper fills that gap with a complete inductive proof under RoPE (Appendix B), providing a theoretical foundation that benefits future work in this area.

The effects of cumulative cache length lU, buffer size lI, attention sinks, and resolution scaling are all analyzed. The offline vs. streaming trade-off curve in Figure 3c effectively situates the method within the broader design space.

- Weakness

 Figure 5 only reports cosine similarity of value caches, without further tracing how this similarity drop affects downstream attention distributions and final predictions. It remains unclear how much similarity degradation is needed to meaningfully hurt performance — the severity of the phenomenon is described qualitatively rather than quantitatively.

The latency analysis in Table 5 is conducted only on StreamingBench (short videos). For long videos such as RVS-Ego/Movie (up to one hour), where query frequency can be considerably higher, the cumulative cost of repeated instant cache construction is not reported. The authors argue that lI=4 keeps overhead acceptable, but no latency curve is provided across varying lI values.

The authors assert in Section 4 that DSCache is orthogonal to compression, pruning, and retrieval-based methods, but no combination experiments are presented to support this claim (beyond the brief StreamingVLM replacement in Appendix D.1). This remains an intuition rather than an empirical finding.

The feature buffer is re-encoded from scratch at every query, meaning the same frames may be encoded multiple times when queries arrive frequently. This is somewhat at odds with the streaming motivation of avoiding redundant computation, yet no degradation strategy or discussion is provided for high query-frequency scenarios.

---

> ### Author Rebuttal · Authors · 2026-03-31
>
> Dear reviewer QsWm
>
> We sincerely thank you for the time and valuable feedback, and appreciate your acknowledgement of our valuable contribution, including the cumulative effect identification, inductive proof,  and thorough analysis. Below we address your questions.
>
> **▶ W1:  Quantitative analysis of similarity degradation vs. performance**
>
> Thanks for the insightful suggestion. The performance drop of the uniform cache baseline reflects the impact of similarity degradation from cumulative effects. To further quantify this, we fix the cache size for inference and vary how the current frame cache is constructed (single-frame, with a feature buffer, or from cumulative cache). Using the single-frame as reference, we measure cache similarity and the resulting performance. As in Tab. A, increasing context length in cumulative cache increases residual accumulation, which reduces cache similarity and degrades performance. This indicates a strong correlation between cache similarity and task performance.
>
> *Tab A. Similarity vs. performance (StreamingBench, Llava-OV-7B)*
> | Setting | | Cos Similarity | Accuracy |
> |----|---|-------|--------|
> | Buffer | | 0.82 | 77.96 |
> | Cumulative Cache |4s| 0.81 | 77.76 |
> |  |8s | 0.79 | 77.20 |
> |  |16s | 0.74 | 76.79 |
> |  |32s | 0.71 | 76.67 |
>
> **▶ W2: Latency analysis for long videos**
>
> To clarify, we report latency at query time, i.e., per-query latency in Tab. 5, when a response is generated and latency is most relevant (the overhead for non-query frames is negligible and comparable to the uniform cache baseline).  Besides, we adopt cache eviction to maintain a fixed-size cache, bounded by a predefined context window (buffer size + cumulative past cache); so per-query latency and memory usage remains constant and does not scale with input length.
>
> Latency mainly comes from instant cache construction at query frames. Tab. B shows larger buffers increase latency. In our setting, a small buffer (e.g., 4 frames) achieves high performance over uniform cache (79.12% vs. 76.92%) while keeping latency manageable (0.30s vs. 0.21s in Tab. 5).
>
> *Tab. B Buffer size(#frames) ablation (LLaVA-OV-7B, StreamingBench)*
> | Buffer Size | Latency | Accuracy |
> |---|-----|------|
> | 2  | 0.26 | 78.74 |
> | 4  | 0.30 | 79.12 |
> | 8  | 0.35 | 79.08 |
> |12 | 0.39 | 79.24  |
>
> To improve efficiency, we propose an approximate design that avoids fully recomputing the instant cache at each query. It maintains an extra periodically refreshed cache for recent frames, enabling reuse of buffer-frame caches while preventing residual accumulation. Tab. C shows that this largely reduces latency, bringing it close to the uniform baseline. Latency also remains stable as buffer size increases(~0.22s up to a 16-frame buffer).  However, this approximation introduces a mild trade-off: periodic resets vary context across frames, slightly affecting performance.
>
> *Tab C. Instant cache approx. (Llava-ov-7B, StreamingBench, a 8-frame buffer).  A large period leads to greater performance drop due to residual accumulation.*
> | Method | Period (s) | Latency (s) | Accuracy |
> |-------|-----|------|------|
> | Uniform Caching | - | 0.21 | 76.92 |
> | DSCache | - | 0.35 | 79.08 |
> | DSCache with approximation | m = 4 | 0.22 | 78.86 |
> | | m = 8 | 0.23 | 78.98 |
> | | m = 16 | 0.22 | 78.48 |
>
> **▶ W3:  Compatibility with KV compression/retrieval**
>
> Thanks for pointing this out. Existing methods[1, 2] focus on cache compression/retrieval over long context. DSCache is complementary, prioritizing accurate recency encoding to better capture current events. In fact, It can be combined with existing methods to improve long-range modeling. We observe further gains when applying these to the cumulative past cache.
>
> *Tab D. Combining DSCache with existing methods(LLaVA-OV-7B)*
> | Method | StreamingBench | OVOBench (Real / Backward / Forward) |
> |-------|-------|----------|
> | DSCache | 79.12 | 57.5 (71.5 / 47.8 / 53.1) |
> | + ReKV [1] | 79.41 | 58.5 (72.4 / 49.1 / 53.8) |
> | + InfiniPot-V [2] | 79.60 | 58.4 (71.8 / 48.9 / 54.7) |
>
> [1] Streaming Video Question-Answering with In-context Video KV-Cache Retrieval, ICLR, 2025
>
> [2] Memory-Constrained KV Cache Compression for Streaming Video Understanding, NeurIPS, 2025
>
> **▶ W4:  Redundancy of buffer re-encoding**
>
> Thanks for raising this concern.  Our implementation follows the sparse-query setting commonly used in streaming video understanding benchmarks. We agree frequent queries may introduce repeated and redundant computation, and our approximation solution in W2 can mitigate this. The periodically refreshed cache enables reuse of recent frame caches across queries without recomputation. We will include more analysis in the revision.

---

> > ### Author Rebuttal · Reviewer_QsWm · 2026-04-02
> >
> > All my concerns are addressed.

---

### Official Review · Reviewer_nd5C · 2026-03-13

**Soundness:** 4
**Presentation:** 4
**Significance:** 3
**Originality:** 3
**Overall Recommendation:** 4
**Confidence:** 4

**Summary:**

The paper operates in the field of streaming video understanding where models must handle unbounded video inputs under strict memory and computation constraints. It proposes DSCache, a training-free mechanism that separates the construction of cumulative past kv caches from on-demand instant caches. This decoupling addresses the identified cumulative effect where residual information from evicted history degrades the encoding quality of recent inputs. The framework was evaluated on multiple streaming VQA benchmarks using LLaVA-OV and Qwen2.5-VL backbones. DSCache achieves state-of-the-art performance, yielding an average 2.5% accuracy gain while supporting infinite sequences via a position-agnostic encoding strategy.

**Compliance With Llm Reviewing Policy:**

Affirmed.

**Key Questions For Authors:**

1. What is the impact of different feature buffer sizes on the trade-off between latency and perception accuracy?
2. How does the system manage position-indexing if the total sequence length grows to several million tokens?
3. Could a hierarchical caching strategy help recover the causal reasoning performance lost in the current decoupling?

**Limitations:**

The authors have adequately discussed the computational trade-off of re-encoding the instant cache and acknowledged the difficulty in long-range reasoning. However, a deeper discussion on potential biases introduced by the position-reindexing (as noted in L1953) would be beneficial.

**Strengths And Weaknesses:**

**Strengths**
1. The authors provide an insightful observation of the cumulative effect in uniform streaming kv caches, backed by cosine similarity analysis between offline and streaming encodings.
2. The inclusion of a formal mathematical proof for the equivalence of position-agnostic encoding in RoPE-based LLMs provides strong theoretical grounding for the method's correctness.
3. The proposed method is training-free and model-agnostic, demonstrating significant performance gains across various competitive open-source backbones.

**Weaknesses**
1. Recomputing the instant cache from the feature buffer introduces a prefilling latency overhead, which, while lower than full recomputation, might still impact throughput in high-fps real-time scenarios.
2. The evaluation shows a slight performance trade-off on subtasks requiring long-range causal reasoning or counting, suggesting that the current decoupling might favor recency over global context coherence.
3. The implementation details for handling independent queries may limit the model's performance in interactive multi-turn dialogues where past query context is crucial.

---

> ### Author Rebuttal · Authors · 2026-03-31
>
> Dear reviewer nd5C
>
> We greatly thank you for your time and insightful comments, and are glad that our observations, theoretical grounding, and design are recognized. We address your questions below.
>
> **▶ W1. Recomputing introduces latency**
>
> Thanks for the concern. Recomputing the instant cache adds latency, which also increases with respect to buffer size (Tab. C). In our setting, a small buffer (e.g., 4 frames) achieves high performance over uniform cache (79.12 vs. 76.92%) while keeping latency manageable (0.30 vs. 0.21s). Meanwhile, latency remains within the input temporal resolution (1 FPS), making it suitable for streaming.
>
> To improve, we propose an approximation that avoids fully recomputation at each query. It maintains a periodically refreshed cache for recent frames, enabling reuse of buffer-frame caches while preventing residual accumulation. This reduces latency (Tab. A) to the uniform baseline level. Latency also remains stable as buffer size increases (~0.22s up to a 16-frame buffer).  However, the approximation incurs a mild trade-off: periodic resets vary context across frames, slightly affecting performance.
>
> *Tab A. Instant cache approx. (Llava-OV-7B, StreamingBench, a 8-frame buffer). A large period hurts performance due to residual accumulation.*
> |Method|Period (s)|Latency (s)|Acc.|
> |--|--|--|--|
> |Uniform Caching|-|0.21|76.92|
> |DSCache|-|0.35|79.08|
> |DSCache with approximation |m=4|0.22|78.86|
> | |m=8|0.23|78.98|
> | |m=16|0.22|78.48|
>
> **▶ W2. Recency vs Global trade-off & Hierarchical caching**
>
> Thanks for the valuable insight.
> - Our method does have a global-recency trade-off. This trade-off is also observed in the literature. Prior methods [1, 2] preserve a recent window while compressing the rest to balance both.
> - DSCache is complementary as it prioritises recency to capture current events. It can be combined with compression/retrieval strategies to improve long-range modeling. We observe further gains when applying such strategies to the cumulative past cache (e.g., backward tracing in Tab. B).
> - As suggested, hierarchical/multi-scale caches are promising and compatible with our work for better capturing global context. We leave this exploration to future work.
>
> *Tab B. Combining DSCache with existing methods(LLaVA-OV-7B)*
> |Method|StreamingBench|OVOBench (Real / Backward / Forward)|
> |--|--|--|
> |DSCache|79.12|57.5 (71.5 / 47.8 / 53.1)|
> |+ ReKV [3]|79.41|58.5 (72.4 / 49.1 / 53.8)|
> |+ InfiniPot-V [1]|79.60|58.4 (71.8 / 48.9 / 54.7)|
>
> [1] Memory-Constrained KV Cache Compression for Streaming Video Understanding, NeurIPS, 2025
>
> [2] Efficient Online Video Understanding via Streaming-Oriented KV Cache and Retrieval, arXiv 2025
>
> [3] Streaming Video Question-Answering with In-context Video KV-Cache Retrieval, ICLR, 2025
>
> **▶ W3. Multi-turn dialogues**
>
> Thanks for raising this. We follow existing benchmarks and baselines, typically restricted to independent query settings. DSCache can extend to interactive settings by maintaining a dedicated text cache for dialogue history alongside the visual cache. At inference, text and visual caches are concatenated with position re-indexing, allowing jointly reasoning over visual and dialogue history. We also include captioning experiments (Suppl. D1) where maintaining textual memory is crucial. We will highlight this in revision.
>
> **▶ Q1. feature buffer size vs. accuracy/latency**
>
> We provide ablations in Fig. 3(b) and detail latency in Tab. C. Smaller buffers preserve fidelity but lack sufficient context; larger buffers increase latency. A moderate size achieves a better balance.
>
> *Tab. C Buffer size(#frames) ablation (LLaVA-OV-7B, StreamingBench)*
> |Buffer Size|Latency|Acc.|
> |--|--|--|
> |2|0.26|78.74|
> |4|0.30|79.12|
> |8|0.35|79.08|
> |12|0.39|79.24|
>
> **▶ Q2. Position-indexing for million tokens**
>
> We adopt cache eviction to maintain a fixed-size cache, bounded by a predefined window (buffer + cumulative past cache); so cache size remains constant and does not scale with input length, avoiding unbounded token growth (e.g., 1M tokens). This enables handling long streams (e.g., ~0.7M tokens for a 1-hour video) and unbounded stream scaling.
>
> For the kept cache, we use position-agnostic encoding with post-hoc positions reindexed from 0 after each eviction. This removes dependence on absolute positions, keeps position bounded, and avoids overflow and the need for extrapolation, ensuring stable operation under long streams.
>
> **▶ Q3. see W2**
>
> **▶ Position-reindexing biases in L1953**
>
> Thanks. The cited line does not correspond to our draft, and we would appreciate clarification on this concern.
>
> Our position-agnostic, reindexed encoding is proposed as standard position-aware caching may fail: cache eviction introduces position discontinuities between system and visual caches, and long streams can cause out-of-distribution or overflow positions. On StreamingBench, we observe a performance drop from 79.12 to 77.84 when using standard position-aware caching.

---

> > ### Author Rebuttal · Reviewer_nd5C · 2026-04-04
> >
> > My concerns have been adequately addressed and I have decided to maintain my original score.

---

### Decision · Program_Chairs · 2026-04-30

**Decision:**

Accept (regular)

**Comment:**

This paper proposes DSCache, a training-free mechanism for streaming video understanding that decouples historical and immediate KV caches to mitigate cumulative residual interference.

All four reviewers recommend a Weak Accept. The committee highlighted the identification of the under-explored "cumulative effect" in streaming caches as highly valuable. Furthermore, the formal mathematical proof for position-agnostic encoding provides a solid theoretical foundation. In addition, the authors successfully addressed major concerns regarding latency overhead and ablation studies. While one reviewer noted the approach feels somewhat incremental (e.g., handling RoPE is an obvious step), the unanimous positive consensus and the strong theoretical and practical contributions justify that this paper is just right above the acceptance threshold.